# Randomized, placebo controlled phase I trial of the safety, pharmacokinetics, pharmacodynamics and acceptability of a 90 day tenofovir plus levonorgestrel vaginal ring used continuously or cyclically in women: The CONRAD 138 study

**Andrea R. Thurman** [1]*, **Vivian Brache**[2], **Leila Cochon**[2], **Louise A. Ouattara**[1], **Neelima Chandra**[1], **Terry Jacot**[1], **Nazita Yousefieh**[1], **Meredith R. Clark**[1], **Melissa Peet**[1], **Homaira Hanif**[1], **Jill L. Schwartz**[1], **Susan Ju**[1], **Mark A. Marzinke**[3], **David W. Erikson**[4], **Urvi Parikh**[5], **Betsy C. Herold**[6], **Raina N. Fichorova**[7], **Elizabeth Tolley**[8], **Gustavo F. Doncel**[1]

**1** CONRAD, Eastern Virginia Medical School, Norfolk and Arlington, VA, United States of America,
**2** Profamilia, Santo Domingo, Dominican Republic, **3** Johns Hopkins University School of Medicine, Baltimore, MD, United States of America, **4** Endocrine Technologies Core (ETC), Oregon National Primate Research Center (ONPRC), Beaverton, OR, United States of America, **5** Department of Medicine, Division of Infectious Diseases and Magee-Womens Research Institute, University of Pittsburgh, Pittsburgh, PA, United States of America, **6** Albert Einstein College of Medicine, Bronx, NY, United States of America, **7** Laboratory of Genital Tract Biology, Brigham and Women's Hospital and Harvard Medical School, Boston, MA, United States of America, **8** Family Health International 360, Research Triangle, NC, United States of America

* thurmaar@evms.edu

## Abstract

Multipurpose prevention technologies (MPTs), which prevent sexually transmitted infection (s) and unintended pregnancy, are highly desirable to women. In this randomized, placebo-controlled, phase I study, women used a placebo or tenofovir (TFV) and levonorgestrel (LNG) intravaginal ring (IVR), either continuously or cyclically (three, 28-day cycles with a 3 day interruption in between each cycle), for 90 days. Sixty-eight women were screened; 47 were randomized to 4 arms: TFV/LNG or placebo IVRs used continuously or cyclically (4:4:1:1). Safety was assessed by adverse events and changes from baseline in mucosal histology and immune mediators. TFV concentrations were evaluated in multiple compartments. LNG concentration was determined in serum. Modeled TFV pharmacodynamic antiviral activity was evaluated in vaginal and rectal fluids and cervicovaginal tissue *ex vivo*. LNG pharmacodynamics was assessed with cervical mucus quality and anovulation. All IVRs were safe with no serious adverse events nor significant changes in genital tract histology, immune cell density or secreted soluble proteins from baseline. Median vaginal fluid TFV concentrations were >500 ng/mg throughout 90d. TFV-diphosphate tissue concentrations exceeded 1,000 fmol/mg within 72hrs of IVR insertion. Mean serum LNG concentrations exceeded 200 pg/mL within 2h of TFV/LNG use, decreasing quickly after IVR removal. Vaginal fluid of women using TFV-containing IVRs had significantly greater inhibitory activity (87–98% versus 10% at baseline; p<0.01) against HIV replication *in vitro*. There was a >10-

**Data Availability Statement:** All relevant data are within the paper and its Supporting Information files.

**Funding:** This study and the clinical development of the TFV/LNG ring were supported by the United States Agency for International Development (USAID) with funds from The U.S. President's Emergency Plan for AIDS Relief (PEPFAR) under Cooperative Agreements (AID-OAA-A-10-00068, AID-OAA-A-14-00010, and AID-OAA-A-14-00011). The Endocrine Technologies Core (ETC) at the Oregon National Primate Research Center (ONPRC). In addition, some of Dr. David W. Erikson's time and effort was supported by National Institutes of Health, (NIH) funding P51 OD011092. Gilead Sciences donated tenofovir active pharmaceutical ingredient (API). The funders had no role in study design, data collection and analysis, decision to publish, or preparation of the manuscript. No author reports any conflict of interest or any competing interests which interfered with the conduct of this study or interpretation of results.

**Competing interests:** The authors have declared that no competing interests exist.

fold reduction in HIV p24 antigen production from ectocervical tissues after TFV/LNG exposure. TFV/LNG IVR users had significantly higher rates of anovulation, lower Insler scores and poorer/abnormal cervical mucus sperm penetration. Most TFV/LNG IVR users reported no change in menstrual cycles or fewer days of and/or lighter bleeding. All IVRs were safe. Active rings delivered high TFV concentrations locally. LNG caused changes in cervical mucus, sperm penetration, and ovulation compatible with contraceptive efficacy.

**Trial registration:** ClinicalTrials.gov #NCT03279120.

## Introduction

Multi-purpose prevention technologies (MPTs), products that offer protection against multiple sexually transmitted infections (STIs) such as herpes simplex virus type 2 (HSV-2) and human immunodeficiency virus type 1 (HIV-1) or STIs and unintended pregnancy, are urgently needed to reduce these global health burdens. Over 37 million people worldwide are infected with HIV-1 and 22.4 million live in sub-Saharan Africa [1]. Almost half of all pregnancies worldwide, estimated to be over 100 million annually, are unintended [2–4]. Unfortunately, highly effective contraceptives (e.g., sterilization, intrauterine devices, hormonal contraception) typically provide no protection against STIs, while barrier methods that protect against STIs (e.g., male or female condoms), have unacceptably high contraceptive failure rates with typical use [5].

CONRAD 128 (ClinicalTrials.gov NCT02235662) was a first-in-woman, randomized, placebo controlled, double blind phase I trial to study the safety, pharmacokinetics (PK), pharmacodynamics (PD), and acceptability of the tenofovir (TFV) intravaginal ring (IVR), the TFV/levonorgestrel (LNG) IVR or placebo IVR worn continuously over approximately 15 days of use [6, 7]. As is often done in first-in-human assessments of investigational new drugs, initial exposure duration was limited to less than the full expected product use duration, but long enough to assess safety with daily use and to establish drug concentrations at steady state. In this initial study, participants inserted the study IVR at the end of menses and removed it prior to the start of the next menses and therefore the impact of the IVR on menstrual bleeding was not assessed [6]. To limit exposure of this investigational product to male partners, sexual abstinence was required during product use [6]. While the CONRAD 128 study was able to establish safety and the initial PK profile and some exploratory PD endpoints, the CONRAD 138 study, described in this manuscript, builds upon and expands these previous data [6, 7].

CONRAD 138, was the first-in-woman study to evaluate the TFV/LNG or placebo IVR over the full 3 months duration using the IVRs in either a continuous or interrupted use regimen (4 treatment/dosing arms). The objectives of the study were to describe the safety (primary objective), acceptability and PK (secondary objectives) and PD (exploratory objective) of the TFV/LNG IVR versus the placebo IVR, over the full 90 days. Because previous data support that some women may want to remove a continuous IVR intermittently [8–12] and that cyclic removal may impact menstrual bleeding, participants were also randomized to use the TFV/LNG or placebo IVR either continuously for 90 days, or in an interrupted, cyclic manner, wearing the IVR for three 28 day cycles, with a 3 day removal in between each cycle. After initial safety was established with the CONRAD 128 study [6], CONRAD 138 expanded the study population to include women with asymptomatic bacterial vaginosis and allowed sexual activity during product use. This allows for the current data to be applicable to a larger population of target users.

## Materials and methods

### Clinical study

The study visits and procedures are summarized in S1 Table. CONRAD A15-138, the ENRICH (Evaluating New Ring Choices) study, was an outpatient, randomized, partially blinded, placebo-controlled, parallel study conducted at the CONRAD Intramural Clinical Research Center at Eastern Virginia Medical School (EVMS) (Norfolk, VA) and PROFAMI-LIA (Santo Domingo, Dominican Republic). The study was approved by the Advarra Institutional Review Board (IRB) (#Pro00022358) and Comisiòn Nacional de Bioetica (#030–2017), respectively, and registered with ClinicalTrials.gov (#NCT03279120).

Written informed consent was obtained from all participants prior to any study procedures. Enrolled participants were healthy, 18–50 years old, had a body mass index (BMI) less than 30 $kg/m^2$, and reported no use of exogenous hormones and regular menstrual cycles. Participants were not at risk of pregnancy due to consistent condom use, sterilization (of the participant or her male sexual partner), or heterosexual abstinence. Women were excluded if they used depot medroxyprogesterone acetate in the last 10 months, were currently breastfeeding, or had a hysterectomy. All women underwent a screening visit (Visit 1, (V1)) to detect the presence of exclusion factors (e.g. symptomatic bacterial vaginosis, and other active current STIs) (S1 Table). Ovulation was confirmed during screening by a luteal phase serum progesterone (P4) level of ≥3.0 ng/mL at visit 2 (V2) (S1 Table). Qualified participants were enrolled and underwent baseline genital tissue sampling in the luteal phase of the menstrual cycle at visit 3 (V3) and were randomized to one of 4 study arms as described below. In the follicular phase of the subsequent menstrual cycle (menstrual cycle day 6 ± 1 day), we obtained additional baseline vaginal and rectal fluid samples and then participants initiated IVR use at visit 4 (V4) in the follicular phase of the menstrual cycle (S1 Table). Cervico-vaginal (CV) and rectal fluid for PK and microbiome analyses were obtained at least monthly (S1 Table). For participants assigned to wear the IVR in a cyclic manner, the IVR was removed at the end of months 1 and 2 (visits 13 (V13) and 22 (V22) respectively), cleaned with 70% isopropyl alcohol, dried, photographed, and kept in a sterile carrying case in the clinic for 3 days and then reinserted at the next visit.

All participants had the IVR removed at visit 31 (V31). At this end of treatment visit, the IVRs were cleaned with 70% isopropyl alcohol, photographed and sent to the laboratory for processing of residual drug concentration (for active IVRs) and objective biomarkers of adherence (for placebo IVRs). Based on self-report and observations of IVR use in the clinic, for the continuous arms, the total number of days with the IVR was determined as (the last day of IVR removal–the first day of IVR insertion + 1)–the total number of days of unintentional IVR removal. The expected number of days with the IVR for continuous use was 90 days. For the interrupted arms, the total number of days with the IVR was determined as (the last day of IVR removal–the first day of IVR insertion +1)–(the total number of days of scheduled and unintentional IVR removal). The expected number of days with the IVR for interrupted use was 84 days.

Adherence was also assessed by objective biomarkers including residual glycerin content and penetrated bioanalytes, for placebo IVR users. Glycerin present in water extracts of IVRs was measured using an enzymatic, colorimetric assay according to manufacturer's instructions (Sigma-Aldrich, St. Louis, MO) as previously reported [13]. In addition, the concentration of bioanalytes that penetrated the placebo IVRs were quantitated using the CBQCA Assay, a fluorescence-based total protein assay (Thermo-Fisher Scientific, Waltham, MA) [13]. The CBQCA reagent reacted with any biological material containing free amine groups in the IVR extracts to generate a fluorescent signal. Because active TFV/LNG IVRs needed to be processed to calculate residual TFV and *in vivo* drug release, the adherence biomarker assessment was

done in placebo IVRs only. Ring residual TFV and *in vivo* TFV release data were also analyzed showing all rings have been used. These data were reported and correlated with the vaginal microbiota and PK in a separate manuscript [14], due to the abundance and complexity of these data.

Post treatment PK was assessed in vaginal tissue and vaginal and rectal fluids at 48 hours, 72 hours, or 5 days post IVR removal (S1 Table). All participants completed an acceptability questionnaire prior to IVR insertion at V3, at the end of month 1 (V13) and the end of treatment (V31). A subset of participants completed an in-depth interview (IDI), conducted by FHI360 via phone at the end of treatment (V31). Acceptability data, other than impact on the menstrual cycle, are being published separately.

### Randomization

At V3 (enrollment), eligible participants were randomized to one of four study arms: continuous TFV/LNG IVR, interrupted TFV/LNG IVR, continuous placebo IVR, or interrupted placebo IVR. We utilized electronic randomization within the Medrio electronic data capture system in a 4:4:1:1 ratio. Participants were randomized to study arm and also received a random time point assignment at V3 for 24, 48, or 72 hours post-IVR insertion sample collection at V5, given the inability to collect all three biopsies from the same woman, every 24 hours, due to safety and feasibility concerns. At Visit 26 or 27 (V26, V27), participants received a random time point assignment for post-IVR removal sample collection (48 hours, 72 hours or 5 days) at the post treatment visit, Visit 32 (V32) (S1 Table).

The randomization scheme was stratified by site and treatment group to maintain balance within each treatment group with respect to the number of participants randomized to each sampling time point. Trial participants, laboratory staff, investigators, and statistical/data analysts were blinded to study treatment and dosing regimen to the extent possible.

### Study product

TFV/LNG and placebo IVRs were manufactured under current good manufacturing practices (cGMP) at Particle Sciences (Bethlehem, PA) using manufacturing processes previously described [15, 16]. The placebo IVR has a similar appearance and dimensions to the TFV/LNG IVR; in lieu of TFV in the hollow reservoir core segment, a pre-gelatinized starch was used as a non-eluting filler. The TFV/LNG IVR is comprised of a hollow hydrophilic polyurethane reservoir sheath with a 55 mm outer diameter; the IVR is filled with a drug-loaded semi-solid core containing ~1.2–1.6 g TFV per IVR, glycerol and water. The TFV/LNG IVR has a 2 cm-long solid hydrophobic polyurethane reservoir segment loaded with 6 mg LNG. The TFV/LNG IVR was designed to release approximately 8–10 mg/day of TFV and 20 µg/day of LNG for 90 days *in vivo*. The IVRs do not require cold chain storage and were stored at room temperature. The mean force (in Newtons) to compress the IVRs to 10% of the diameter (F10) was 2.40 (range 1.78–2.99) for the placebo IVR and 1.86 (range 1.40–2.30) for the TFV/LNG IVR; these values are within the range of other commercially available IVRs [17].

### Clinical and sub-clinical safety assessments

**Adverse events.** Adverse events (AEs) were the primary safety measure, along with any changes at the end of treatment in safety laboratory (complete blood count, fasting lipids, and comprehensive chemistry panel) measurements from baseline. We monitored AEs at each study visit, starting with the enrollment visit, (V3 S1 Table) and thus all reported AEs appeared during treatment and are considered treatment-emergent adverse events (TEAEs), whether they are related or not to the interventions. We graded each AE for severity, (mild, moderate,

severe, potentially life threatening or death) using the DAIDS tables for grading the severity of adverse events (http://rsc.tech-res.com/clinical-research-sites/safety-reporting/daids-grading-tables) and relationship to study product or study procedures (graded as related versus not related). This process included assignment of relatedness by the clinical site PI and confirmation of classification by the Sponsor's Medical Director. All AEs were coded with the appropriate MDR code for the clinical study report and for data presentation. A suspected adverse reaction was defined as any AE for which there was a reasonable possibility that the study product caused the event. An AE was considered "serious" if, in the view of either the investigator or the sponsor, it resulted in death, was immediately life threatening, required an unplanned in patient hospitalization or resulted in persistent or significant disability or incapacity. Finally, we considered a AE to be unexpected if it was not listed in the investigator brochure, was not listed at the specificity or severity that has been observed; or, was not consistent with the risk information described in the general investigational plan.

In response to findings of genital ulcers with the use of another IVR containing tenofovir disoproxil fumarate (TDF) [18], we added additional 4 quadrant inspection pelvic exams with a lighted speculum, so that each participant had at least 13 pelvic exams for safety assessment after IVR insertion (S1 Table). We contacted participants 1 to 2 weeks after final genital sampling to ask about AEs experienced and medications taken since the last visit.

**Density and phenotype of immune cells in ectocervical tissue.** One ectocervical biopsy at pre-insertion baseline (V3) and at the end of treatment (V31) was placed in 10% neutral buffered formalin for 24–48 hours, transferred to an embedding cassette, and processed as per our immunohistochemistry protocol [19] for immune markers at the Profamilia site (S1 Table). For the EVMS site, the ectocervical biopsy was placed in an empty cryovial and transferred to the laboratory. In the laboratory, the biopsy was cut in to two pieces, if possible, with one portion being processed as above in formalin for detection of CD45, CD3, CD8, and HLA-DR. The other piece, if available, was processed by cryopreservation for detection of CD4 and CCR5. Additional methodology is contained within the S1 File.

**Secreted soluble proteins from the cervico-vaginal mucosa.** At visits 4 and 29, a cervico-vaginal fluid lavage (CVL) was collected in 4ml normal saline after speculum insertion and lavage of the cervical fornices and vaginal walls, avoiding spraying directly into the cervical os. The concentration of soluble proteins in the supernatant was then measured by multiplex electrochemiluminescence assay and ELISA, using procedures established under accreditation by the College of American Pathologists [6]. See S1 File for additional information.

## Pharmacokinetic assessments

**Tenofovir pharmacokinetics.** TFV concentrations in plasma, rectal fluid, vaginal fluid, and vaginal tissue biopsies were collected throughout product use at multiple time points (S1 Table) and determined via a previously described liquid chromatographic-tandem mass spectrometric (LC-MS/MS) analysis [20]. Vaginal fluid was collected on Dacron swabs, and rectal fluid was collected on Merocell sponges; TFV concentrations were determined from matrix and collection device-specific calibration standards. See S1 File for assay lower limits and additional PK parameter details.

**Levonorgestrel pharmacokinetics and sex hormone binding globulin.** Serum LNG concentrations were obtained at multiple visits throughout the study (S1 Table) and were measured with a Shimadzu Nexera-LCMS-8050 liquid chromatography-tandem triple quadrupole mass spectrometry (LC-MS/MS) platform (Shimadzu Scientific, Kyoto, Japan) using a modification of a previously published method [21]. See S1 File for additional PK parameter details.

## Pharmacodynamic assessments

**Anti-human immunodeficiency virus activity in cervico-vaginal fluid lavage and rectal fluids.** A CVL was collected at baseline (prior to IVR insertion), month 1, and in month 3 near the end of treatment (V29) (S1 Table). An aliquot of the supernatant was processed for evaluation of anti-HIV activity. Rectal fluid was collected using Merocell sponges at V4 and V29 (S1 Table). Rectal fluid was eluted from rectal sponges in 400 μL of PBS. Briefly, TZM-bl cells were incubated with CVL supernatant or rectal fluid at a 1:16 final dilution. Anti-HIV activity was determined by infecting cells with 3000 $TCID_{50}$ of HIV-1Ba-L and measuring luminescence after 48 hours using Bright-Glo$^{TM}$ (Promega Corporation, Madison, WI). Percent inhibition of HIV-1$_{Ba-L}$ was determined based on deviations from the HIV-1-only control as previously described [22–24].

**Anti-herpes simplex virus type-2 activity in vaginal and rectal fluids.** Vaginal fluid was obtained by direct aspirate from the posterior vaginal fornix using a 2.5 mL vaginal fluid aspirator at baseline pre-insertion (V4) and at the end of treatment (V29) (S1 Table) (CarTika Medical, Maple Grove, MN) and immediately transferred to a cryovial and stored at -80 ˚C until processing. Thawed samples were diluted with 400 μL of normal saline and centrifuged at 2,000 rpm for 7 minutes at 4˚C. Rectal fluid was eluted from rectal sponges in 400 μL of normal saline. The activity of the fluid against HSV-2 was measured by plaque reduction assay as previously described [25]. Virus was mixed 1:1 with cervico-vaginal fluid (CVF), rectal fluid or control buffer before infecting Vero cells in duplicate and the percent change in number of plaques quantified [25].

We also tested anti-HSV-2 activity from Dacron swabs obtained after IVR use (at days 10, 21, 32, 42, 63, 84) (S1 Table) in a subset of participants, using a different assay method with HEC1a cells. In this second analysis, vaginal swabs were placed in 200μl of HEC1A media for 5–10 minutes. Then the swabs were centrifuged to extract the CVF/media from the swab. HEC1A cells were seeded in 48 well plates and the following day were treated for total of 6 hours with the CVF/media in duplicates. In the last hour of incubation cells were infected with about $1X10^{-3}$ multiplicity of infection (MOI) per well HSV-2(G) isolate (ATCC, VR-734) for an hour. The treatment and inoculum were removed and fresh media was added and cells were incubated for 5 days. HSV-2 DNA was evaluated by a quantitative RT-PCR amplification of cell culture supernatants on day 5 using SYBR-green (Roche, Basel, Switzerland). Supernatants (6 μl) were amplified using the forward primer 5′- TCGCCAGCACAAACTCAT -3′ and the reverse primer 5′- CCACCGACCTCAAGTACAAC -3′ targeting glyprotein B of HSV-g isolate. The amount of HSV-2 DNA in all treated wells was compared to untreated control. Results are presented as relative HSV-2 DNA expression of each well to untreated control, with a reduction of DNA expression equal to higher inhibition.

**p24 antigen production by tissue biopsies infected ex vivo with HIV-1$_{BaL}$.** One vaginal and one ectocervical biopsy were collected at baseline and one ectocervical biopsy was collected at the end of treatment (S1 Table). Tissues were placed in cryovials filled with chilled RPMI 1640 media (Life Technologies, Carlsbad, CA) containing 10% fetal bovine serum (ATCC, Manassas, VA) and 100 U/ml penicillin and 100 μg/ml streptomycin (Thermo Fisher Scientific, Waltham, MA) (cRPMI). The CV biopsies were exposed to HIV-1$_{BaL}$ ($5x10^4$ $TCID_{50}$/mL), within 30 minutes of collection, in presence of Interleukin-2 human (hIL-2) (Roche Diagnostics GmbH) at a final concentration of 100 U/mL. CV biopsies were washed 2–3 hours after virus exposure and then cultured in cRPMI media (500 μL) containing IL-2 for 21 days. Data was analyzed as previously reported [26]. See S1 File for additional details.

**Levonorgestrel pharmacodynamic surrogate assessment.** The potential for contraceptive efficacy was modeled by several surrogates, including ovulation during IVR use,

categorized as any serum P4 of $\geq 3$ ng/mL during a 28 day period, as previously described [27]. To detect ovulation and time cervical mucus assessments, participants returned approximately twice weekly after IVR initiation, to have serum estradiol (E2) and P4 measured. We performed a cervical mucus check for Insler score [28] and sperm penetration assay (modified slide test) [29–33] at the next regularly scheduled visit, usually within 48 hours, if the serum E2 was between 75–150 pg/mL. If the serum E2 was over 150 pg/mL, indicating imminent follicular development, we brought the participant in for a cervical mucus check within 24 hours. If the serum E2 did not reach 75 pg/mL before the end of each month (i.e. days 28, 59 and 90), we collected cervical mucus at the end of the month 1 (V13) month 2 (V22) or month 3 (V31) visits (S1 Table). See S1 File.

## Acceptability assessments

We used a good clinical practices (GCP) and code of federal regulations (CFR) compliant, professionally administered electronic survey system for participants to answer questions about any menstrual cycle changes with product use, opinions on product characteristics, impact on the menstrual cycle, and the acceptability of the study products. Participants answered survey questions in private areas of both clinics at baseline prior to IVR insertion (V3), after one month of use and at the end of treatment. The impact of the IVRs on the participant's menstrual cycle is reported in this manuscript.

## Sample size and statistical analyses

Sample size for this phase I study was primarily based on the size of similar studies and feasibility, although statistical differences were considered. We ultimately had 10 participants in the placebo arm, which was the number of placebo IVR assigned participants in our first in woman study of the rings [6]. In this previous study, we were able to demonstrate statistical differences in contraceptive and anti-viral surrogates between the placebo and active IVRs [6]. SAS® software version 9.4 (SAS Institute, Inc., Cary, NC, USA) was used for analysis. The primary objective of the present study, safety, was compared using the Fisher's exact test whenever possible for the reporting of AEs between treatment groups.

The randomized population (RP) included all randomized participants, who received their product randomization assignment and dosing assignment at enrollment (S1 Table). The treated population (TP) is a subset of RP and consisted of all randomized participants with any amount of IVR use. The evaluable population (EP) is a subset of TP and consisted of all randomized participants with any IVR use and contributing at least some follow-up safety or PK/PD data. The EP will be the primary analysis population for study objectives.

All data sets were examined for normality using the PROC UNIVARIATE procedure in SAS and examining the histogram, Q plot and the Shapiro Wilk statistic. All continuous variables were not normally distributed and therefore appropriate non-parametric statistical testing was then performed.

The proportion of participants reporting AEs were compared based on treatment group using a Fisher's exact test whenever feasible or for low cell size or a Chi square test. The Wilcoxon signed rank sum test was used to compare paired changes from baseline, pre-insertion (visit 4) to month 1 (HIV inhibition by CVL) and or baseline pre-insertion to month 3 (end of treatment) for safety endpoints to the end of treatment values (e.g. density of tissue lymphocytes and concentrations of secreted soluble proteins) as these data are not normally distributed. Similarly, the Kruskall Wallis test was used to compare safety endpoints among the 4 independent treatment/dosing groups at the end of treatment.

PK analysis included descriptive summaries by sampling time point of: TFV concentrations in plasma, vaginal fluid, rectal fluid, and vaginal tissue; TFV-diphosphate (TFV-DP) concentrations in vaginal tissue and LNG and SHBG concentrations in serum. For summaries of concentrations, measurements below the level of quantification (BLQ) were imputed as $0.5^*$ lower limit of quantification (LLOQ).

PD endpoints including anti-HIV-1 activity in vaginal and rectal fluids and anti-HSV-2 activity in vaginal and rectal fluids, p24 antigen production from CV tissues after *ex vivo* infection with HIV-1$_{BaL}$ (at the EVMS site only), HSV-2 infectivity in vaginal tissue (HSV-2 DNA fold change on day 12, cumulatively, and AUC) (at the EVMS site only), and cervical mucus quality (Insler) score and sperm penetration assay results were summarized using descriptive statistics by time points. For anti-HIV and p24 antigen production endpoints, an ANOVA model was used to test if the mean values of TFV/LNG (continuous) differed from placebo (continuous), TFV/LNG (interrupted) differed from placebo (interrupted), and pooled TFV/LNG differed from pooled placebo. Baseline was defined as the pre-insertion measurement. Within group changes from baseline were compared using paired t-tests (Wilcoxon signed rank sum test for non-parametric data). Categorical PD endpoints of qualitative measurement of TFV and sperm migration were summarized by time point using number and percent of participants in each category. Proportions of participants in each category were compared by chi square or Fisher exact test, depending on the cell size.

PK versus PD correlations were performed using a Spearman correlation coefficient, as these data were not normally distributed. The log10 normalized data were used for investigating any linear relationship between PK and PD variables. For the anti-HSV2 post hoc analyses of data obtained using the HEC1a cell assay, we used receiver operator curve (ROC) analysis, to categorize the anti-HSV2 activity for each sample as inhibitory/positive or non-inhibitory/negative. Once categorized in this manner, the anti-HSV2 data from placebo IVR users versus TFVLNG exposed visits versus TFV/LNG IVR removal visits was compared with Fisher's exact test.

Nominal p-values are reported, unadjusted for multiple analyses and considered significant at $p < 0.05$.

## Results

### Study population

The first participant was enrolled in October 2017, and the last participant completed the study in December 2018. As summarized in Fig 1, 68 participants were screened and 47 enrolled in the study (37 randomized to the TFV/LNG IVR and 10 randomized to the placebo IVR). The most common reasons for screen failures were serum progesterone $< 3$ ng/mL in the baseline luteal phase, indicative of anovulation at V2 (n = 10), and an STI at screening (n = 6). All women in the placebo arms completed the study. Seven participants in the TFV/LNG arms did not complete the study due to personal reasons (Fig 1). No participant was withdrawn from the study due to AEs. The demographic data of the randomized participants are summarized in Table 1. One participant in the interrupted TFV/LNG arm was not included in the enrolled population due to protocol violations.

### Expulsions and adherence

Based on participant diary responses, mean adherence (proportion of days IVR was used of the total expected days of use) with continuous treatment was 93.26% and 99.78% for TFV/LNG and placebo, respectively. Eight and 3 participants were 100% compliant, 6 and 2 participants were $\geq 80\%$ to $< 100\%$ compliant, and 2 and 0 participants were $< 80\%$ compliant for

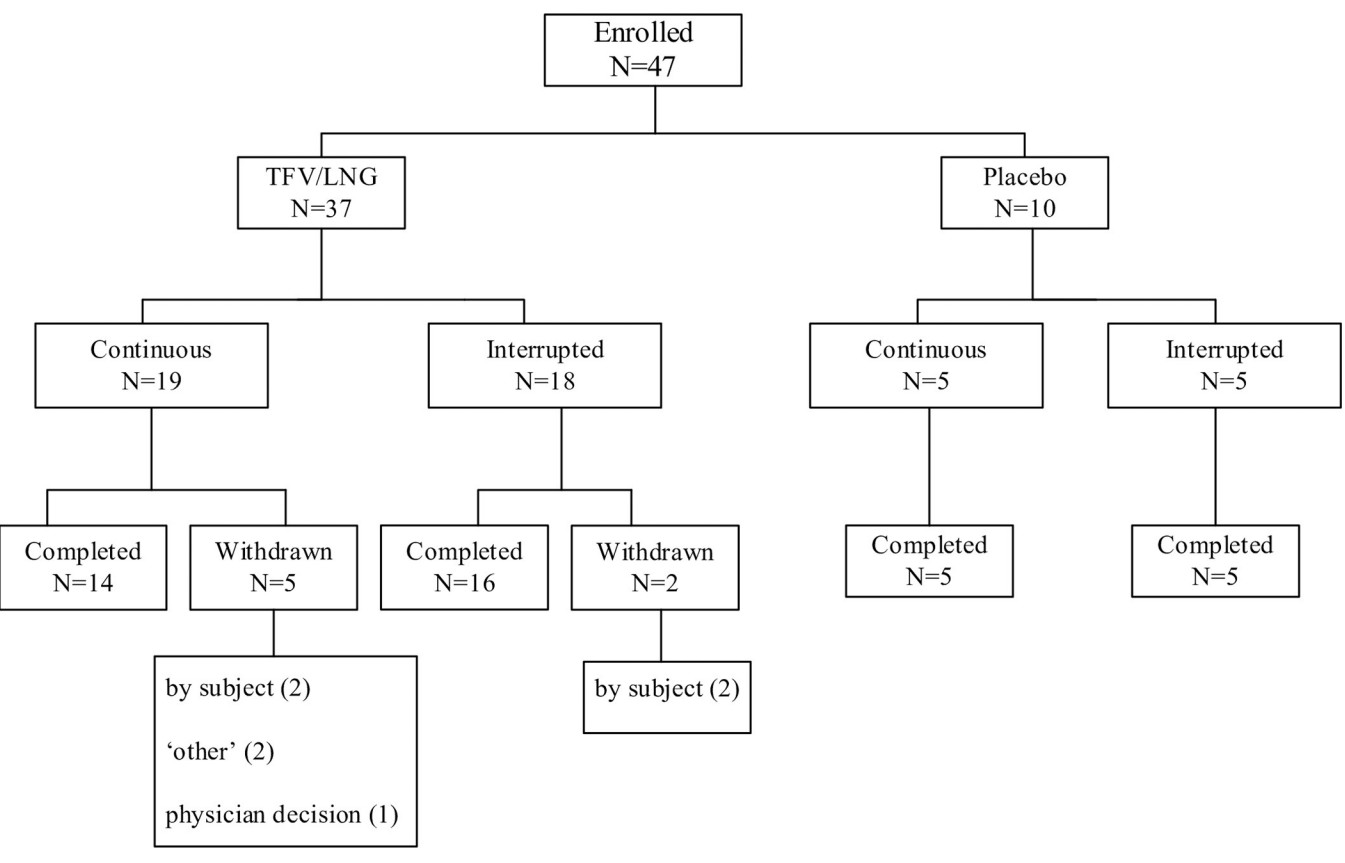

**Fig 1. Disposition of participants.**

continuous TFV/LNG and placebo ring use, respectively. With interrupted treatment, mean compliance (proportion of days IVR was used of the total expected days of use) was 93.85% and 100% for TFV/LNG and placebo, respectively. All but 3 participants were 100% compliant with the treatment regimen: 1 participant assigned to TFV/LNG was $\geq$ 80%—< 100% compliant, and 2 participants were < 80% compliant, both in TFV/LNG.

Analysis of the placebo IVRs showed a median residual glycerin of 0.01 and 0.02 mM for the placebo cyclic and continuous groups respectively (p = 0.34) Post use, the placebo IVRs contained a median of 5,790 and 15,010 μg/mL of penetrated bioanalytes in the cyclic and continuous cohorts respectively (p = 0.17).

Participants reported no spontaneous ring expulsions during treatment.

## Safety endpoints

**Duration of use and adverse events.** There were no serious adverse events (SAEs), unexpected AEs or deaths. Overall, the incidence of participants experiencing AEs was generally similar across treatments. As shown in Table 2, there were no statistically significant differences (calculated using exact methods) in the proportion of participants in each treatment and dosing group reporting various types of AEs. The most common AEs ($\geq$ 5% overall) were vaginal discharge, viral upper respiratory tract infection, influenza, malaise, nausea, back pain, abdominal pain lower, diarrhea, and decreased hemoglobin.

Most AEs were assessed as mild or moderate and the combined incidence of reports of mild or moderate AEs was not different from the 4 study groups (Fisher exact p value = 0.64).

**Table 1.  Demographic and baseline characteristics treated population.**

| | TFV/LNG (Continuous) | TFV/LNG (Interrupted) | Placebo (Continuous) | Placebo (Interrupted) | Total TFV/ LNG | Total Placebo |
|---|---|---|---|---|---|---|
| | (n = 18) | (n = 18) | (n = 5) | (n = 5) | (n = 36) | (n = 10) |
| **Age (years)** | | | | | | |
| Mean (SD) | 37.8 (5.2) | 36.7 (6.8) | 33.6 (4.2) | 33.6 (6.8) | 37.3 (6.0) | 33.6 (5.3) |
| **Body Mass Index (kg/m2)** | | | | | | |
| Mean (Standard Deviation, SD) | 27.0 (2.7) | 25.1 (2.7) | 26.6 (2.5) | 23.8 (2.5) | 26.0 (2.8) | 25.2 (2.8) |
| **Ethnicity** | | | | | | |
| Hispanic/Latina | 12 (66.7%) | 9 (50.0%) | 3 (60.0%) | 4 (80.0%) | 21 (58.3%) | 7 (70.0%) |
| Not Hispanic/Latina | 6 (33.3%) | 9 (50.0%) | 2 (40.0%) | 1 (20.0%) | 15 (41.7%) | 3 (30.0%) |
| **Race** | | | | | | |
| American Indian or Alaska Native | 0 | 1 (5.6%) | 0 | 0 | 1 (2.8%) | 0 |
| Asian | 0 | 1 (5.6%) | 0 | 0 | 1 (2.8%) | 0 |
| Mixed Race | 11 (61.1%) | 8 (44.4%) | 2 (40%) | 3 (60%) | 19 (52.8%) | 5 (50%) |
| Black/African American | 3 (16.7%) | 4 (22.2%) | 2 (40%) | 1 (20%) | 7 (19.4%) | 3 (30%) |
| White | 4 (22.2%) | 4 (22.2%) | 1 (20%) | 1 (20%) | 8 (22.2%) | 2 (20%) |
| **Education (years)** | | | | | | |
| Mean (SD) | 11.4 (3.9) | 12.4 (3.6) | 12.0 (4.0) | 12.6 (2.3) | 11.9 (3.7) | 12.3 (3.1) |
| **Contraceptive Method Used in Study** | | | | | | |
| Sterilization of either partner | 15 (83.3%) | 13 (72.2%) | 3 (60.0%) | 4 (80.0%) | 28 (77.8%) | 7 (70.0) |
| Abstinence | 2 (11.1%) | 1 (5.6%) | 0 | 1 (20.0%) | 3 (8.3%) | 1 (10.0%) |
| Non-spermicidal condoms | 1 (5.6%) | 4 (22.2%) | 2 (40.0%) | 0 | 5 (13.9%) | 2 (20.0%) |
| **Study Partner Status** | | | | | | |
| Living with study partner | 13 (72.2%) | 13 (72.2%) | 4 (80.0%) | 2 (40.0%) | 26 (72.2%) | 6 (60.0%) |
| Not living with study partner | 2 (11.1%) | 4 (22.2%) | 0 | 0 | 6 (16.7%) | 0 |
| No study partner | 3 (16.7%) | 1 (5.6%) | 1 (20.0%) | 3 (60.0%) | 4 (11.1%) | 4 (40.0%) |
| **Was the participant ever pregnant?** | | | | | | |
| Yes | 18 (100%) | 16 (88.9%) | 5 (100%) | 5 (100%) | 34 (94.4%) | 10 (100%) |
| No | 0 | 2 (11.1%) | 0 | 0 | 2 (5.6%) | 0 |

The Grade 3 AEs included amoebiasis, decreased hemoglobin, and diarrhea reported by one participant each in continuous TFV/LNG treatment, and post procedural hemorrhage reported by one participant in interrupted placebo treatment. None of the Grade 3 AEs were considered related to study treatment or study procedure (Table 2).

AEs assessed by the investigator as related to study treatment or study procedure was also not statistically different based on randomization to treatment or dosing group (Table 2).

One AE, in the continuous TFV/LNG arm, a 1 cm. vaginal epithelial disruption was considered related to study drug use and was detected at visit 18 (approximately 45 days of IVR use) and led to dose interruption. This mild disruption resolved completely within 10 days. Two unrelated AEs, decreased hemoglobin in a continuous TFV/LNG participant and dysmenorrhea in an interrupted TFV/LNG participant, led to discontinuation of study drug.

The mean duration of treatment in days and mean (SD) total TFV (mg) and LNG (µg) received during the study, by treatment and dosing regimen is detailed in Table 2. Two participants in the continuous TFV/LNG treatment had missing IVR removal data and were excluded from exposure analysis (the calculation of duration of exposure) but included in total dose received calculations.

**Table 2. Summary of duration of use, total dose received and adverse events.**

| | TFV/LNG (Continuous) (N = 18) | TFV/LNG (Interrupted) (N = 18) | Placebo (Continuous) (N = 5) | Placebo (Interrupted) (N = 5) | Fisher exact P value |
|---|---|---|---|---|---|
| Total Number of AEs | 41 | 46 | 7 | 14 | NA |
| Total Number of SAEs | 0 | 0 | 0 | 0 | NA |
| Number (%) of Subjects Reporting at Least One: | | | | | |
| AE | 15 (83.3%) | 15 (83.3%) | 3 (60.0%) | 5 (100%) | 0.57 |
| AE by Severity [1] | | | | | |
| Grade 1: Mild | 8 (44.4%) | 10 (55.6%) | 1 (20.0%) | 1 (20.0%) | 0.43 |
| Grade 2: Moderate | 4 (22.2%) | 5 (27.8%) | 2 (40.0%) | 3 (60.0%) | 0.39 |
| Grade 3: Severe | 3 (16.7%) | 0 (0.0%) | 0 (0.0%) | 1 (20.0%) | 0.29 |
| AE by Relationship to Study Treatment [2] | | | | | |
| Not Related | 8 (44.4%) | 12 (66.7%) | 1 (20.0%) | 5 (100%) | 0.09 |
| Related | 7 (38.9%) | 3 (16.7%) | 2 (40.0%) | 0 (0.0%) | |
| Related Grade $\geq$ 3 | 0 (0.0%) | 0 (0.0%) | 0 (0.0%) | 0 (0.0%) | NA |
| AE by Relationship to Study Procedure [3] | | | | | |
| Not Related | 14 (77.8%) | 15 (83.3%) | 3 (60.0%) | 4 (80.0%) | 0.38 |
| Related | 1 (5.6%) | 0 (0.0%) | 0 (0.0%) | 1 (20.0%) | |
| AE Leading to: | | | | | |
| Dose Interruption | 1 (5.6%) | 0 (0.0%) | 0 (0.0%) | 0 (0.0%) | 1.00 |
| Discontinuation of Study Drug | 1 (5.6%) | 1 (5.6%) | 0 (0.0%) | 0 (0.0%) | 1.00 |
| Duration of Use and Total Dose Received | | | | | |
| Mean (SD) Duration of IVR Use (Days) | 83.9 (15.9) | 78.8 (23.3) | 89.8 (2.6) | 87.6 (1.5) | |
| Mean (SD) Total dose of TFV received (mg) | 855.3 (332.6) | 808.3 (418.9) | NA | NA | |
| Mean (SD) Total dose of LNG received (μg) | 1,637.2 (491.0) | 1,661.7 (493.1) | NA | NA | |

TEAEs are AEs reported or observed during treatment whether they were considered related to treatment or not.

[1] Subjects reporting more than one adverse event are counted only once using the highest severity.

[2] Subjects reporting more than one adverse event are counted only once using the closest relationship to study drug (ie, related or unrelated).

[3] Subjects reporting more than one adverse event are counted only once using the closest relationship to study procedure (ie, related or unrelated).

P values for proportion of participants reporting AEs per dosing group calculated with Fisher's exact test.

Mean values for hematology and serum chemistry parameters were within normal ranges at baseline and end of treatment. No clinically meaningful or dose-related mean changes from baseline were observed.

**Ectocervical immune cells and epithelium.** No statistically significant changes from baseline in the density and phenotype of ectocervical mucosal immune cells and HIV-1 target cells were observed across treatment groups and dosing regimens. In addition, there were no differences in immune cell population phenotypes at the end of treatment based on treatment or dosing regimen (Table 3). In a subset of cryopreserved samples, collected at the EVMS site (n = 12), there were no significant differences in CD4 and CCR5 cell density between TFV/LNG and placebo users (S2 Table). Comparison of CD4 and CCR5 data, however, is limited due to the small number of samples that could be portioned for cryopreservation.

**Secreted soluble proteins from cervico-vaginal mucosa.** In paired comparisons, there were no significant changes from baseline at the end of treatment in any of the soluble markers of innate mucosal immunity and inflammatory response (all paired p values > 0.05). There were also no differences between the drug and dosing regimen cohorts in the soluble protein

**Table 3. Immune cells in ectocervical tissue at the end of treatment (visit 31) based on treatment and dosing assignment.**

| | TFVLNG continuous | | | | TFVLNG Interrupted | | | | Placebo Continuous | | | | Placebo Interrupted | | | | P |
|---|---|---|---|---|---|---|---|---|---|---|---|---|---|---|---|---|---|---|
| | N | Median | IQR 25 | IQR 75 | N | Median | IQR 25 | IQR 75 | N | Median | IQR 25 | IQR 75 | N | Median | IQR 25 | IQR 75 | |
| Histology | | | | | | | | | | | | | | | | | |
| Epithelial Thickness (μm) | 13 | 240 | 206.7 | 268 | 15 | 298 | 223.3 | 378.3 | 5 | 236.7 | 180 | 306.7 | 4 | 258.5 | 191.8 | 372.5 | 0.32 |
| Number Cell Layers | 13 | 17.4 | 16.3 | 19.5 | 15 | 21.7 | 19.7 | 23.3 | 5 | 16.2 | 13.3 | 20.5 | 4 | 17.3 | 14.1 | 24.2 | 0.23 |
| Cells in the Ectocervical Epithelium (cells/mm2) | | | | | | | | | | | | | | | | | |
| CD45 | 13 | 107.9 | 90.2 | 119.7 | 15 | 67.3 | 54.3 | 115.3 | 5 | 116.2 | 67.6 | 148.5 | 4 | 101.9 | 65.1 | 143.3 | 0.36 |
| CD3 | 13 | 67.9 | 57.3 | 80.2 | 15 | 47.9 | 33.3 | 77.3 | 5 | 80.8 | 47.4 | 122.9 | 4 | 69.9 | 49.3 | 90.5 | 0.32 |
| CD8 | 13 | 43.6 | 36.7 | 61.7 | 15 | 30.4 | 10.7 | 58.5 | 5 | 47.1 | 32 | 84.5 | 4 | 40.6 | 27.2 | 59 | 0.27 |
| HLADR | 13 | 36.7 | 30.3 | 46.2 | 15 | 36.5 | 30.8 | 50.2 | 5 | 65.7 | 20.5 | 78.7 | 4 | 30.7 | 25.7 | 50.3 | 0.82 |
| Cells in the Ectocervical Lamina Propria (cells/mm2) | | | | | | | | | | | | | | | | | |
| CD45 | 13 | 72 | 64 | 148 | 13 | 86.4 | 52 | 144 | 5 | 93.3 | 72 | 112 | 4 | 94 | 70 | 169.6 | 0.96 |
| CD3 | 13 | 38.4 | 36 | 106.7 | 13 | 48 | 32 | 82.7 | 5 | 56 | 38.4 | 80 | 4 | 49.3 | 36 | 108.5 | 0.97 |
| CD8 | 13 | 22.4 | 16 | 52 | 13 | 24 | 12 | 48 | 5 | 26.7 | 22.4 | 44.8 | 4 | 28 | 20 | 60.8 | 0.92 |
| HLADR | 13 | 42.7 | 28 | 48 | 13 | 48 | 40 | 64 | 5 | 44.8 | 32 | 50.7 | 4 | 43.3 | 32 | 44.7 | 0.62 |

IQR = Interquartile range

* p values determined using Kruskall Wallis test.

marker profiles at the end of treatment (all p values > 0.12, Table 4). Overall, the CVL immune marker profiles did not indicate a shift towards a pro-inflammatory or immuno-suppressed state.

## Tenofovir pharmacokinetic assessments

**Tenofovir in plasma.** Overall TFV plasma concentrations were very low (mean range 1.16–11.49 ng/mL) in both active treatment dosing groups throughout the study. Pre-insertion

**Table 4. Comparison of independent group soluble proteins at end of treatment.**

| Variable (pg/mL/mg total protein) | TFVLNG Continuous (n = 15) | | | TFVLNG Interrupted (n = 16) | | | Placebo Continuous (n = 5) | | | Placebo Interrupted (n = 5) | | | P value |
|---|---|---|---|---|---|---|---|---|---|---|---|---|---|
| | Median | IQR 25 | IQR 75 | Median | IQR 25 | IQR 75 | Median | IQR 25 | IQR 75 | Median | IQR 25 | IQR 75 | |
| Inflammatory | | | | | | | | | | | | | |
| IL_1RA (x $10^5$) | 2.4 | 1.5 | 6.8 | 2.3 | 1.4 | 5.4 | 2.7 | 1.9 | 3.5 | 1.6 | 1.4 | 3.5 | 0.84 |
| IL_1α | 837.3 | 335.9 | 3805.6 | 760.8 | 460 | 3249 | 1406.2 | 849.2 | 5505.6 | 1329.9 | 328.9 | 2641 | 0.90 |
| IL_6 | 17.6 | 4.7 | 45.6 | 17.5 | 6.1 | 86.6 | 6.5 | 0.9 | 30.4 | 22.9 | 19 | 25.6 | 0.81 |
| IL_8 | 1535.6 | 560 | 3624 | 2811.5 | 586.2 | 4788.2 | 165.9 | 132.7 | 5764.8 | 1698.6 | 1034.2 | 3293.6 | 0.68 |
| IP_10 | 25.8 | 8.1 | 85.9 | 22.4 | 8.7 | 52.2 | 11.1 | 9.9 | 14 | 18.1 | 11.9 | 30.2 | 0.84 |
| GM_CSF | 0.2 | 0 | 2.1 | 0.2 | 0.1 | 0.9 | 0 | 0 | 0.3 | 0.4 | 0.1 | 0.6 | 0.51 |
| MIP_1a | 9.5 | 5.9 | 14.4 | 8.8 | 2.5 | 13.3 | 4.2 | 1 | 6.6 | 8.4 | 8.1 | 13.3 | 0.36 |
| RANTES | 13.5 | 1.3 | 148.4 | 5 | 0.6 | 11 | 1.8 | 1 | 5.1 | 16.9 | 2.7 | 26.4 | 0.57 |
| TNF_α | 1.7 | 0.4 | 5.4 | 1 | 0.4 | 11 | 0.9 | 0.2 | 2.1 | 1.1 | 0.4 | 6.7 | 0.83 |
| Anti-Inflammatory or Anti-Microbial | | | | | | | | | | | | | |
| IL_10 | 0.2 | 0.1 | 0.3 | 0.2 | 0.1 | 0.9 | 0.1 | 0.1 | 0.5 | 0.5 | 0.2 | 2.5 | 0.5 |
| BD2 (x $10^5$) | 0.3 | 0.1 | 6.2 | 0.4 | 0.1 | 1.1 | 8.1 | 0.7 | 8.1 | 1.0 | 0.3 | 2.8 | 1.0 |
| SLPI (x $10^5$) | 3.8 | 1.1 | 5.4 | 2.3 | 1.1 | 4.7 | 3.4 | 2.5 | 5.8 | 2.4 | 1.1 | 6.3 | 2.4 |

* p values determined using Kruskall Wallis test, IQR = Interquartile Range.

plasma concentrations (BLQ) were reached within 48 hours of IVR removal in both TFV/LNG IVR dosing groups. Calculated PK parameters for TFV in plasma are displayed in S3 Table.

**Tenofovir in vaginal and rectal fluids.** In the continuous treatment cohort, TFV in vaginal fluid was detectable within 2 hours of IVR insertion and gradually reached high levels (1,000 ng/mg) at 48 hours post-insertion. TFV levels in CVF remained high throughout day 63 of use (mean TFV 2760 ng/mg), declining in samples taken on days 73 and 84 (end of treatment, EOT) (mean TFV 1102 ng/mg). Within 48 hours of IVR removal, mean TFV concentrations decreased to those reported at 2 hours post insertion (mean TFV 5 ng/mg) (Fig 2A).

A similar pattern was observed for interrupted treatment (Fig 2B), except for the first month re-insertion visit (visit 14) and the second month re-insertion visit (visit 23) where mean TFV concentrations decreased significantly compared to continuous treatment following IVR removal (at Visit 14, 3780 and 143 ng/mg in continuous and interrupted treatment, respectively, p < 0.01; and at Visit 23, 2760 and 256 ng/mg in continuous and interrupted treatment, respectively, p < 0.01) (Fig 2B). Following reinsertion of the IVR, mean vaginal fluid TFV concentrations with interrupted treatment increased to those observed during continuous treatment; at day 73 of use, the mean TFV vaginal fluid concentration in the interrupted arm (2627 ng/mg) was significantly higher than in the continuous arm (1526 ng/mg) (p = 0.03). Calculated PK parameters for TFV in vaginal fluid are displayed in S3 Table.

TFV concentrations in rectal fluid were very low (< 10 ng/mg) (S3 Table). In continuous treatment, TFV was detectable in rectal fluid 24 hours post-insertion, then increased slightly through day 84 (mean range over treatment 0.13–2.06 ng/mg), and then decreased by 72 hours post-removal (range 0.0–0.2 ng/mg). A similar pattern was observed for interrupted treatment (medians range over treatment 0.50–6.29 ng/mg), although with more fluctuation.

**Tenofovir and tenofovir-diphosphate in vaginal tissue.** PK parameters for TFV and TFV-DP in vaginal tissue are displayed in S3 Table. In continuous treatment, TFV was detectable in vaginal tissue 24 hours post-insertion (median 28.6 ng/mg), reaching steady state soon after and maintaining high concentrations through the EOT (median 27.3 ng/mg) before decreasing below post-insertion concentrations at all post IVR removal time points (range 0.16–12.0 ng/mg). A similar pattern was observed for interrupted treatment. There was no statistically significant dosing regimen effect at any time point (p ≥ 0.24) and therefore median (IQR) TFV tissue concentrations among continuous (n = 14) and cyclic dosing (n = 16) regimen users are presented together (Fig 3A). There were non-significant differences among low TFV tissue concentrations at the three time-points post-ring removal. This is likely due to the fact that, for safety reasons, post IVR removal PK tissue biopsies were not taken at each time point from all participants, but rather smaller subsets of the whole had PK tissue biopsies taken at each time point post removal (see randomization section above).

In continuous treatment, median levels of TFV-DP, the active TFV metabolite, exceeded 100 fmol/mg and 1000 fmol/mg by 24 and 72 hours post-insertion, respectively, remaining high through the end of treatment (median 514 fmol/mg) and even up to 5 days post removal (median 440 fmol/mg). A generally similar pattern was observed for interrupted treatment, although with more variability. There was no statistically significant dosing regimen effect at any time point (all p values ≥ 0.25) and therefore median TFV-DP tissue concentration data for the two dosing cohorts (n = 20) are displayed together (Fig 3B).

## Tenofovir pharmacodynamic assessments

**Anti-viral activity against human immunodeficiency virus in vaginal fluids.** There were no significant differences among the 4 treatment/dosing cohorts in the baseline HIV inhibition of the CVL supernatant, prior to IVR insertion (p = 0.49). As outlined in Table 5,

### a: TFV (ng/mg) in Vaginal Fluid Continuous Dosing

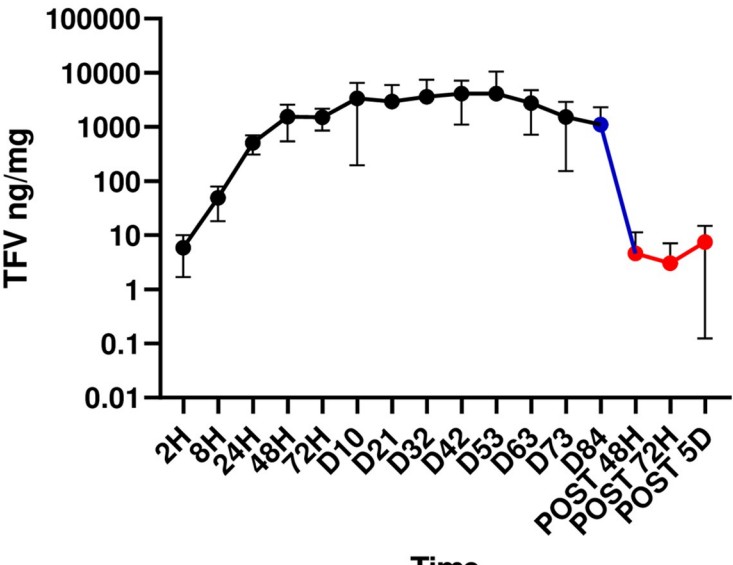

**Black = IVR treatment**
**Blue = End of Treatment**
**Red = Post IVR removal**

### b: TFV (ng/mg) in Vaginal Fluid Interrupted dosing

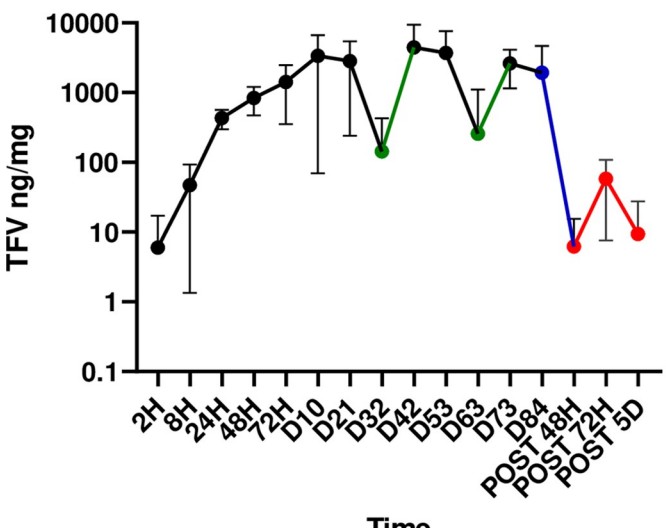

**Black = IVR treatment**
**Green = Post 3 day removal**
**Blue = End of Treatment**
**Red = Post IVR removal**

**Fig 2.** a. Median (IQR) TFV (ng/mg) in vaginal fluid continuous dosing. b. Median (IQR) TFV (ng/mg) in vaginal fluid interrupted dosing. Black = IVR treatment, Green = post 3 day removal, Blue = end of treatment, Red = post IVR removal.

there were no significant changes from baseline in the anti-HIV activity of the CVL among placebo IVR users (p values = 1.00). TFV/LNG IVR users experienced a significant increase in HIV inhibitory activity of the CVL at month 1 compared to baseline (p values < 0.01) and at the end of treatment compared to baseline (p values < 0.01) (Table 5, Fig 4A).

At month 1 and month 3 of ring use, the inhibitory activity of the CVL among TFV/LNG containing IVR users was significantly higher compared to placebo IVR users (both time point p values < 0.01). Among active IVR users, there was also no significant effect of dosing regimen in the HIV inhibitory activity of the CVL at month 1 (V11) (p = 0.96) or near the end of treatment (V29) (p = 0.71). HIV inhibition in CVF was directly correlated with TFV CVF concentrations (Spearman coefficient R = 0.80; p<0.001) (Fig 4B).

**Anti-viral activity against human immunodeficiency virus in rectal fluids.** The baseline mean values for percent HIV inhibition by rectal fluids were similar compared to the respective placebo group for both continuous TFV/LNG and interrupted TFV/LNG treatment. The mean percent changes from baseline to the end of treatment were not statistically significant compared to placebo (p = 0.87 [TFV/LNG in continuous treatment], p = 0.60 [TFV/LNG in both dosing regimens], and p = 0.55 [TFV/LNG in interrupted treatment]). Consistent with low rectal mucosal concentrations of TFV, discussed above, mean percent changes from baseline within the treatment arms were not statistically significant for percent inhibition (p > 0.05).

**Human immunodeficiency virus infection of tissue biopsies infected *ex vivo* with HIV-1$_{BaL}$.** Although there was a clear, several fold decrease in HIV-1 p24 production in tissues exposed to TFV/LNG IVR versus baseline pre-insertion tissues or placebo IVR exposed tissues,

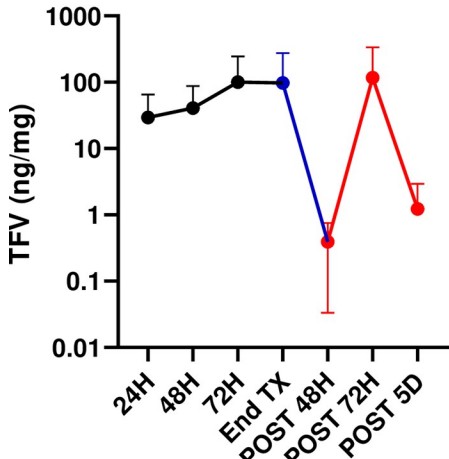

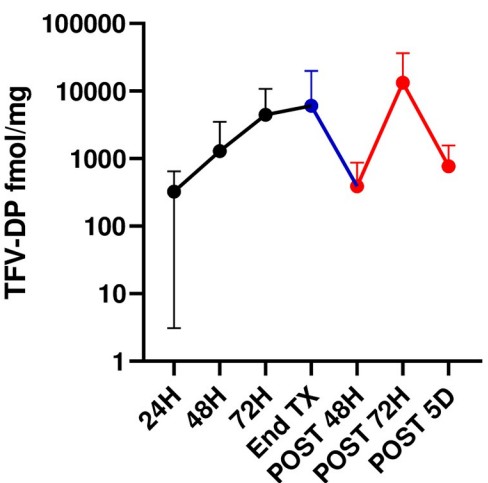

**Fig 3.** a. Median (IQR) tenofovir concentration (ng/mg) in vaginal tissue. b. Median (IQR) TFV-DP concentration (ng/mg) in vaginal tissue. Black = visit 5 (24, 48 or 72 hours post IVR insertion), Blue line = end of treatment; Red = post IVR removal.

**Table 5. *In vitro* HIV inhibitory activity of the CVL supernatant for each treatment and dosing regimen.**

| Variable | Pre-Insertion Baseline (% HIV Inhibition) | | | Month 1 (% HIV Inhibition) | | | | End of Treatment (% HIV Inhibition) | | | | P value |
|---|---|---|---|---|---|---|---|---|---|---|---|---|
| | N | Median | IQR 25 | IQR 75 | N | Median | IQR 25 | IQR 75 | N | Median | IQR 25 | IQR 75 | |
| TFVLNG IVR Continuous | 18 | 9.47 | -12.44 | 22.82 | 16 | 97.26 | 94.32 | 99.19 | 15 | 86.59 | 35.01 | 98.62 | < 0.01 |
| TFVLNG IVR Interrupted | 18 | 11.57 | 0.77 | 21.61 | 17 | 97.89 | 95.61 | 99.14 | 16 | 93.60 | 66.10 | 96.97 | < 0.01 |
| Placebo IVR Continuous | 5 | 0.44 | -15.26 | 7.40 | 5 | -6.94 | -18.75 | 32.34 | 5 | -35.37 | -49.52 | 29.47 | 1.00 |
| Placebo IVR Interrupted | 5 | 4.39 | -7.60 | 13.80 | 5 | 9.45 | -38.29 | 37.10 | 5 | -5.25 | -20.68 | 4.99 | 1.00 |

* p values determined using Wilcoxon signed rank sum test to compare changes from baseline at month 1 and from baseline to month 3. Highest p value reported in table for each row.

overall these changes were not statistically significant, likely due the inherent variability of the assay [26]. In a post hoc analysis, we showed that p24 antigen production at baseline from ectocervical tissues and paired vaginal tissues was not different (all p values > 0.50). Furthermore, baseline p24 production was not different among treatment groups. Similarly, we found that p24 antigen production from ectocervical or vaginal tissues was similar post treatment for TFV/LNG continuous and TFV/LNG interrupted arms (all p values > 0.11). Therefore, as shown in Table 6, in a *post hoc* analysis, we compared p24 antigen production between baseline samples from all groups (all treatments and dosing regimens at baseline plus placebo samples) versus end of treatment samples in the TFV/LNG IVR groups and, although not statistically significant, found a >10 fold decrease in HIV-1 p24 antigen production from tissues post exposure to TFV (median AUCs are 1074 and 102 (below the limit of detection) for tissues not exposed and exposed to TFV, respectively; p = 0.20). HIV p24 tissue production (area under the curve, 21 day tissue culture) was inversely correlated with TFV concentrations in tissue (Spearman coefficient R = -0.27; p<0.04).

**Anti-viral activity against herpes simplex virus type 2 in vaginal and rectal fluids.** Using the Vero cell plaque assay, the mean percent inhibition of HSV-2 by diluted vaginal fluid was not statistically different among the 4 independent dosing groups (p values > 0.10, data shown in S4 Table) at baseline, month 1 and month 3. There was a significant increase in HSV-2 inhibition from baseline to end of treatment in the TFV/LNG continuous arm (p = 0.02, S4 Table), but no significant changes were seen from baseline to the end of treatment for the TFV/LNG interrupted regimen and both placebo groups (p values > 0.15, S4 Table). Similarly, there was no significant change in percent inhibition of HSV-2 plaque formation when virus was mixed with rectal fluid *in vitro*.

In a post hoc assessment of anti-HSV-2 activity minimizing CVF dilution by using HEC1a cells and HSV-2 DNA PCR as endpoint (Fig 5A), we found a statistically significant (p<0.0001) difference between the inhibitory activity of combined placebo samples and those of continuous TFV/LNG IVR users, in combination with samples from interrupted IVR users when the ring was *in situ* (median HSV-2 inhibition = 4.4 vs 43.3, respectively, Fig 5A). In contrast, samples from TFV/LNG IVR interrupted dosing users when the IVR had been out for 3 days did not show significant differences from placebo (Fig 5A).

We determined a ROC threshold of 19.36 fold DNA inhibition to categorize sample anti-HSV activity as inhibitory/positive or non-inhibitory/negative. There was a statistically significant (Fisher's exact test p<0.0001) difference associating TFV-releasing rings in situ versus placebo IVRs versus visits when the TFV/LNG IVR had been removed with anti-HSV activity in CVF. To verify this association, we found a statistically significant linear correlation (R = 0.54, p < 0.01) between the TFV vaginal fluid concentration and HSV-2 DNA inhibition, reinforcing the specificity of the antiviral effect (Fig 5B).

## a. TFV/LNG (Continous & Cyclic) Vaginal Fluid HIV Inhibition

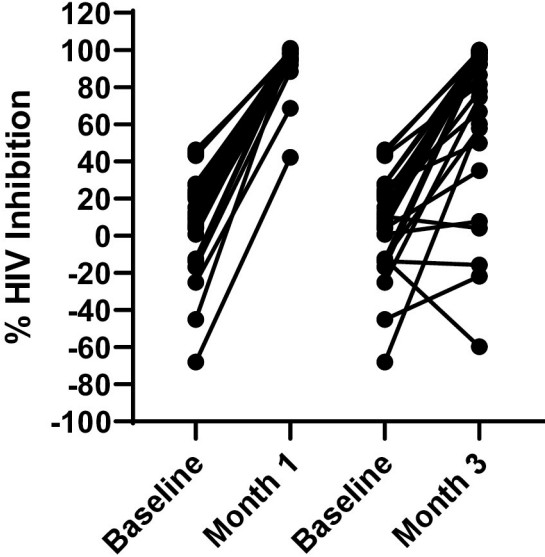

**p < 0.01 change in pre-insertion to months 1 and 3**

## b. Correlation between [TFV] and Anti-HIV Activity in CVF TFV/LNG IVRs

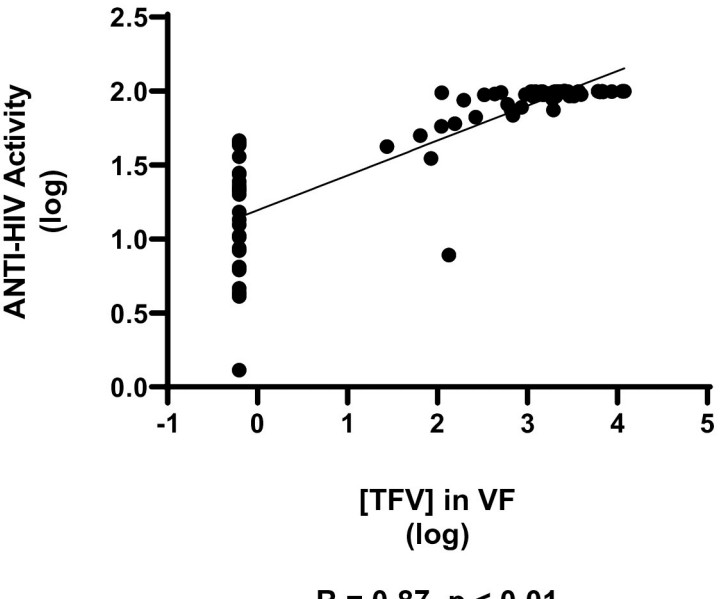

**R = 0.87, p < 0.01**

**Fig 4.** a. Inhibition of HIV growth *in vitro* by vaginal fluids of TFV/LNG IVR users (continuous and cyclic dosing regimens combined) at baseline and 1 and 3 months after use. b. Correlation between [TFV] and Anti-HIV activity in CVF TFV/LNG IVRs.

### LNG PK assessments

**LNG in serum.** In continuous treatment, mean levels of LNG increased following IVR insertion to above 200 pg/mL by 4 hours post-insertion and remained above 200 pg/mL through visit 17 (month 2, approximately day 42 of IVR use). At the end of treatment mean

**Table 6. p24 antigen production from ectocervical and vaginal tissue biopsies.**

| TX Group and P24 variable | P24 antigen production (pg/mL) from placebo and baseline ectocervical and vaginal tissue biopsies | | | | P24 antigen production (pg/mL) from ectocervical tissue biopsies after 3 months of TFV/LNG IVR use | | | | P value |
|---|---|---|---|---|---|---|---|---|---|
| | N | Median | IQR 25 | IQR 75 | N | Median | IQR 25 | IQR 75 | |
| p24 SOFT | 44 | 300 | 6 | 2613 | 12 | 6 | 6 | 2323 | 0.55 |
| p24 Area Under Curve | 44 | 1074 | 102 | 14,690 | 12 | 102 | 102 | 5481 | 0.23 |
| p24 Cumulative | 44 | 384 | 36 | 5371 | 12 | 36 | 36 | 2897 | 0.55 |
| p24 Day 21 | 44 | 140 | 6 | 2388 | 12 | 6 | 6 | 2323 | 1.00 |

* p values calculated using a Kruskall Wallis test.

IQR = Interquartile Range.

LNG concentration in serum was 155.5 (46.8) pg/mL for continuous TFV/LNG IVR users (Fig 6). With interrupted treatment (Fig 6), mean levels of LNG increased following IVR insertion to above 200 pg/mL by 2 hours post-insertion and remained above 200 pg/mL through the end of treatment while the IVR was inserted. Consistent with periods of IVR non-use, mean LNG serum concentrations at Visit 14 (approximately day 32, after first 3 day interruption) and Visit 23 (approximately day 63, after 2nd 3 day interruption) decreased to near pre insertion levels following removal of the IVR, and were significantly lower at these re-insertion visits compared to continuous treatment at both time points.

Although there were no statistically significant differences in SHBG between the continuous and interrupted treatment arms at any time point, a trend of greater mean values from the interrupted group versus the continuous treatment group after Visit 14 was observed ($p \geq 0.0514$).

## LNG PD assessment

We pre-specified the surrogate PD criteria for modeling pregnancy protection as the presence of one or more of the following surrogates of contraceptive efficacy (Table 7): anovulation (all serum progesterone measurements during the month of $< 3$ ng/mL), an Insler score of $\leq 10$, indicating poor cervical mucus quality, and abnormal sperm penetration into the CM. More participants in TFV/LNG arms had cervical mucus scores $\leq 10$, abnormal or poor sperm penetration assays and anovulatory cycles over 3 months compared to placebo arms (Table 7). All

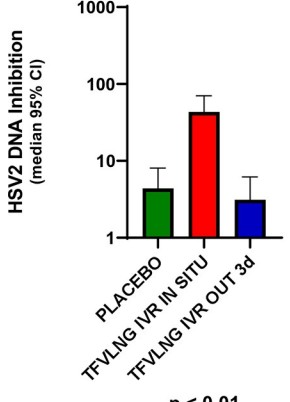
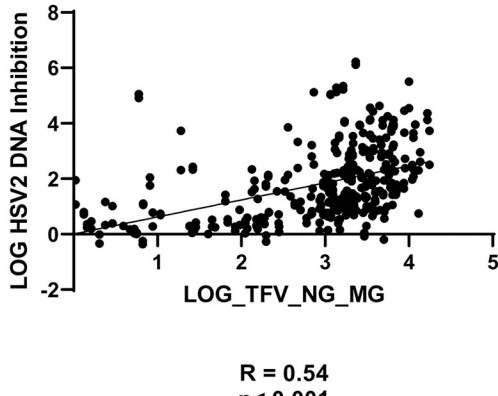

**Fig 5.** a. Median (IQR) HSV2 DNA inhibition. b. Correlation HSV2 inhibition versus TFV in vaginal fluid (TFVLNG IVRs).

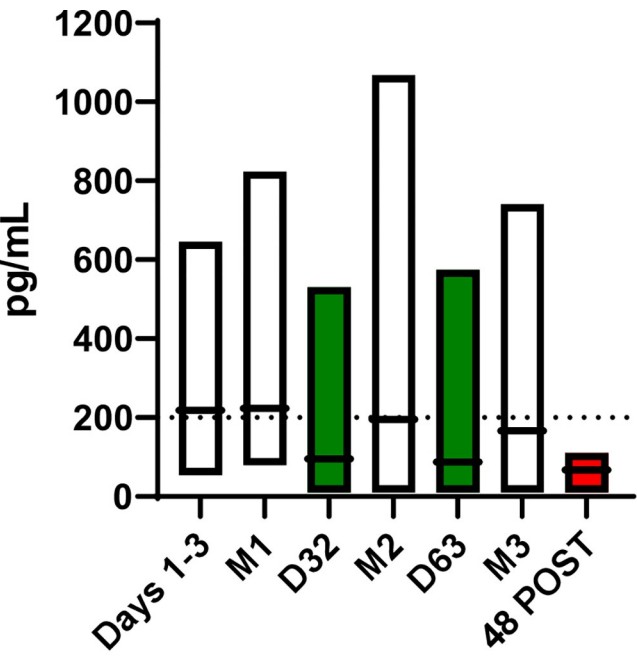

**Days or Months after IVR insertion**
**Green Bars contain trough values after 3d removals**
**Red = 48 hours post removal**

**Fig 6. Serum LNG (pg/mL) in TFV/LNG continuous IVR Users (n = 18) and in TFV/LNG interrupted IVR users IVR users (n = 17).**

TFV/LNG IVR users had at least one surrogate of contraceptive efficacy. All Placebo IVR users ovulated monthly for 3 months.

### Acceptability: Effect on menstrual cycle

Among all participants, 43.2% and 45% reported no changes in their menstrual cycle at month 1 and the end of treatment, respectively. Although placebo IVR users had the highest proportion of participants reporting no change to their menstrual cycle, there were no statistically significant differences in the proportion of women in the placebo groups or active dosing regimens reporting any particular menstrual cycle change (all p values > 0.22) (Table 8). The majority of TFV/LNG IVR users, whether the active IVR was used continuously or cyclically, reported either no change in their menstrual cycle or fewer days and/or lighter bleeding (Table 8). Between 14.3%– 25.0% of active IVR users reported at either month 1 or the end of treatment that they experienced heavier and or more days of menstrual bleeding, but this was not different from the proportion of placebo IVR users reporting this change at either month 1 or month 3 (p values > 0.22) (Table 8).

### Discussion

The TFV/LNG IVR was safe and well tolerated. The IVR delivered high local tissue concentrations of TFV and TFV-DP. Altogether TFV and LNG PK and PD data support potential for contraceptive and HIV prevention efficacy for this MPT product. There were no significant differences in the proportion of participants randomized to one of the four treatment and dosing groups in their reports of various levels of AEs and the relatedness of these AEs to the

**Table 7. Pharmacodynamic surrogates of levonorgestrel activity.**

| Period and Variable | TFV/LNG Continuous | TFV/LNG Interrupted | Placebo Continuous | Placebo Interrupted |
|---|---|---|---|---|
| Cervical Mucus Insler Score (normal > 10) | | | | |
| Month 1 (N) | 18 | 17 | 5 | 5 |
| Mean (SD) | 6.9 (3.3) | 6.5 (3.6) | 11.0 (3.3) | 12.8 (1.1) |
| Median | 6.0 | 5.0 | 11.0 | 12.0 |
| % with score > 10 | 4 (22.2%) | 3 (17.6%) | 3 (60.0%) | 5 (100%) |
| Month 2 (N) | 16 | 16 | 5 | 5 |
| Mean (SD) | 7.0 (2.8) | 7.5 (3.7) | 10.2 (3.8) | 12.2 (2.5) |
| Median | 7.5 | 5.5 | 13.0 | 13.0 |
| % with score > 10 | 1 (6.3%) | 6 (37.5%) | 3 (60.0%) | 4 (80.0%) |
| Month 3 (N) | 14 | 16 | 5 | 5 |
| Mean (SD) | 6.7 (2.4) | 8.3 (3.4) | 11.4 (2.7) | 12.2 (1.9) |
| Median | 7.0 | 7.0 | 12.0 | 13.0 |
| % with score > 10 | 1 (7.1%) | 6 (37.5%) | 4 (80.0%) | 4 (80.0%) |
| Sperm Migration Assay | | | | |
| Month 1 (N) | 13 | 12 | 5 | 4 |
| Normal | 1 (5.6%) | 2 (11.8%) | 5 (100%) | 3 (60.0%) |
| Month 2 (N) | 15 | 11 | 4 | 5 |
| Normal | 4 (22.2%) | 4 (23.5%) | 4 (80.0%) | 5 (100%) |
| Month 3 (N) | 14 | 15 | 5 | 5 |
| Normal | 4 (22.2%) | 4 (23.5%) | (80.0%) | 5 (100%) |
| Monthly Ovulation by Serum Progesterone | | | | |
| Month 1 (N) | 18 | 17 | 5 | 5 |
| Ovulation | 7 (38.9%) | 8 (47.1%) | 5 (100%) | 5 (100%) |
| Month 2 (N) | 16 | 16 | 5 | 5 |
| Ovulation | 10 (62.5%) | 8 (50.0%) | 5 (100%) | 5 (100.0%) |
| Month 3 (N) | 14 | 16 | 5 | 5 |
| Ovulation | 10 (71.4%) | 11 (68.8%) | 5 (100%) | 5 (100%) |

study product or study procedures. Just prior to the start of this trial, a 90-day phase I study of an IVR releasing TDF, a prodrug of TFV, in sexually active women was prematurely terminated due to grade 1 genital ulcers in 8 of 12 women randomized to the TDF IVR [18]. In response to these events, we added 13 additional, four-quadrant cervico-vaginal exams to the

**Table 8. Reported impact of IVR use on menstruation (percent of participants per group reporting impact, participants may report more than one menstrual related impact).**

| N (% of IVR users at time point) | Regimen and Agent | | | | | | Chi square p value responses at month 1 | Chi square p value responses at month 3 |
|---|---|---|---|---|---|---|---|---|
| | TFV/LNG Continuous | | TFV/LNG Interrupted | | Placebo Continuous and Interrupted | | | |
| | Month 1 (n = 18) | End of Treatment (n = 16) | Month 1 (n = 16) | End of Treatment (n = 14) | Month 1 (n = 10) | End of Treatment (n = 10) | | |
| No change in cycle | 8 (44.4) | 6 (37.5) | 6 (37.5) | 6 (42.9) | 5 (50.0) | 6 (60.0) | 0.81 | 0.52 |
| Fewer days and/or lighter bleeding | 6 (33.3) | 5 (31.3) | 6 (37.5) | 5 (35.7) | 4 (40.0) | 2 (20.0) | 0.93 | 0.70 |
| Heavier and/or more days | 4 (22.2) | 4 (25.0) | 4 (25.0) | 2 (14.3) | 1 (10.0) | 0 (0.0) | 0.63 | 0.22 |
| More irregular menstrual bleeding | 3 (16.7) | 4 (25.0) | 2 (12.5) | 2 (14.3) | 1 (10.0) | 1 (10.0) | 0.87 | 0.57 |

study, in order to increase safety monitoring. One participant, randomized to the TFV/LNG IVR, who was not sexually active, presented with a 1 cm epithelial disruption of her left vaginal epithelium after approximately 45 days of IVR use. We removed her IVR and her epithelial disruption completely resolved within 10 days without additional treatment. The epithelial disruption was located in the mid left vaginal sidewall, not near the vaginal fornices and ectocervix where the IVR normally contacts the genital mucosa and where the majority of ulcers were found with the TDF IVR [18]. Despite this, we classified this AE as product related, since the participant was not sexually active and denied use of any other intravaginal products. We observed no other epithelial findings among any of the other 46 enrolled participants. Previously, an *in vitro* modeling study raised concern that TFV or more potent pro-drugs might impair wound healing in the female reproductive tract [34]. It was thought that this could potentially uncover safety signals when topical anti-retrovirals were used in sexually active women, as consensual vaginal intercourse can cause micro-trauma to the CV mucosa [35]. Unlike our previous first-in-woman trial [6], women enrolled in this study were permitted to engage in sexual activity. Although we did not collect data on the frequency of sexual intercourse, ninety percent of enrolled participants were sexually active. We found no evidence of mucosal disruption or ulcers associated with IVR use and sexual activity. Our data support that the TFV/LNG IVR, when used by healthy, sexually active women, does not cause genital ulceration or any other significant AEs.

Twelve of 18 TFV/LNG IVR continuous dosing users reported mild or moderate AEs versus 15 of 18 TFV/LNG IVR cyclic dosing users. While this difference was statistically significant, we do not believe that it is clinically relevant, as the difference is small and AEs were mostly mild and unrelated to product use.

Severe (Grade 3) AEs were observed only in participants in continuous TFV/LNG treatment and interrupted placebo treatment. The Grade 3 AEs included amoebiasis, decreased hemoglobin, and diarrhea reported by one participant each in continuous TFV/LNG treatment, and post procedural hemorrhage reported by one participant in interrupted placebo treatment. None of the Grade 3 AEs were considered related to study treatment or study procedure (Table 2).

AEs assessed by the investigator as related to study treatment occurred at similar frequencies within continuous treatment (38.9% and 40.0% in TFV/LNG and placebo, respectively). In interrupted treatment, the incidence of related AEs was higher in TFV/LNG than in placebo arms (16.7% and 0%, respectively); none of these related AEs occurred in more than one participant. Related AEs were more frequent in continuous TFV/LNG (38.9%) than interrupted TFV/LNG (16.7%) treatment.

We found that the majority of active IVR users reported no change in their menses or lighter bleeding/less bleeding days with IVR use. Our data are consistent with the previously tested 20 μg/day LNG IVR, which found that a quarter of the IVR users experienced some type of menstrual bleeding irregularity, but only 15% discontinued due to that reason [36, 37]. Breakthrough bleeding is one of the most common causes of discontinuation for progestin only contraceptives [38]. We hypothesized that breakthrough bleeding would occur more frequently with higher doses of progestins which, systemically, are associated with higher rates of anovulation [39]. In this study, we included the cyclic arm to determine the impact of IVR removals on menstrual bleeding patterns and because previous data from several contraceptive and HIV-1 prevention IVR trials support that some women prefer to remove the IVR with intercourse or menstrual bleeding [8–12]. Demonstration of acceptable menstrual bleeding patterns among the majority of active IVR users in this study is highly encouraging. Furthermore, our data support that intermittent removal of the TFV/LNG IVR does not appear to offer any improvement in menstrual bleeding patterns over continuous IVR use.

Although limited to the placebo IVR cohort (n = 10), analysis of the residual glycerin content indicated that the IVRs were used for 30 or more days based on previous data [13]. In placebo IVRs glycerin is fully expelled within approximately 1 month of use and therefore this assay is a good indicator of poor or non-adherence [13]. Median residual glycerin in used placebo IVRs from both dosing cohorts in this study (84–90 days of use) was similar to the median residual glycerin in placebo IVRs after 30 days of use in our previous adherence biomarker validation study [13]. The combined median levels of penetrated bioanalytes into used placebo IVRs from both dosing cohorts combined in this study was significantly higher than median concentrations measured among women using the placebo IVRs for 1 month in our previous validation study of these objective markers [13], indicating use for longer than 1 month.

Our previous first-in-woman study of the TFV and TFV/LNG IVRs was the first to report the impact of topical LNG on mucosal safety endpoints [6], such as CV immune cell populations and local cytokine and chemokine production. Unlike both depot medroxyprogesterone acetate and oral, systemic LNG combined with ethinyl estradiol [40, 41], the locally applied LNG combined with TFV in the IVR did not cause any significant changes in soluble CV biomarkers with up to 3 months of use. In the TDF ring study, the investigators noted several changes in CV cytokines and chemokines when sexually active women used the ring [42]. In the current study, we report no changes in these endpoints from baseline with the placebo or TFV/LNG IVRs.

An essential mechanism of the microdose LNG is the local effect on CM, which has been characterized with the LNG intrauterine system [43], but has not been correlated with plasma LNG concentrations measured from systemically administered contraceptives (reviewed in [44]). Steady state serum levels of LNG, measured by radio-immunoassay (RIA) in previous studies of the 20 μg/day LNG IVR, ranged from 187–682 pg/mL [39, 45, 46]. Serum concentrations of LNG in systemic, LNG contraceptive implant (Norplant, Jadelle) users generally exceed a mean of 200–300 pg/mL [47–50], as measured by RIA. This range is often considered the systemic benchmark for contraceptive efficacy (reviewed in [44]). In our first study of the TFV/LNG IVR [6], we were able to use RIA to provide comparisons to previous RIA determined benchmark data from systemic, LNG based contraceptives [47–50]. In this study, serum LNG concentrations were measured with LC-MS/MS, the current gold standard for LNG PK assessments [21], which does not have an established comparison to past benchmarks measured by RIA. However, our data support that the TFV/LNG IVR delivers levels of LNG consistent with previously tested LNG IVRs from the WHO [39, 45, 46] and in the range of effective systemic LNG implants [47–50] and current, micro-dose LNG contraceptive intrauterine systems [51, 52]. Not surprisingly, we observed a significant decrease in serum LNG during IVR removal in the interrupted dosing arm, as compared to the continuous dosing arm. It is unknown whether these decreases in systemic LNG concentrations would reduce contraceptive efficacy with cyclic, interrupted use, but given that cyclic use does not improve bleeding profiles over continuous use, these data would support continuous use of the IVR as the preferred contraceptive modality. The impact of short drops in LNG plasma concentration on local contraceptive mechanisms such as those related to cervical mucus rheology, however, has not been clearly established.

In our previous study [6], we timed the only cervical mucus check based on the urinary luteinizing hormone (LH) surge. In this study, we used twice weekly E2 and P4 serum testing to more precisely define ovulation and time the multiple cervical mucus checks. This was more accurate as all participants in the placebo arms showed evidence of monthly ovulation. The anovulation rate among TFV/LNG IVR users in this study was similar to or higher than that seen in previous studies of LNG micro-dose contraceptive IVRs [45, 53–55].

The maximum plasma concentration ($C_{max}$) and plasma steady state concentrations of TFV were well within the low range expected from topical, as opposed to oral, dosing of TFV [56–58]. Surrogates of protection against HIV-1 for TFV containing topical microbicides are still modeled by concentrations of TFV in the CV aspirate and its association with HIV sero-conversion in the CAPRISA 004 study, which utilized pre and post coital TFV vaginal gel, not a sustained-release topical TFV source, as provided by the TFV or TFV/LNG IVRs [59]. A CV aspirate TFV concentration over 100 ng/mL conferred an estimated 65% protection against HIV-1 acquisition, while a CV aspirate TFV concentration of over 1,000 ng/mL provided an estimated 76% protection against HIV-1 [59, 60]. Recently, TFV concentrations in vaginal fluid have been expressed as nanograms per milligram (ng/mg) of vaginal fluid, to more accurately normalize the concentration; thus the TFV concentrations in this study are given as ng/mg. We found that continuous TFV/LNG IVR users maintained high vaginal fluid TFV concentrations, above 1000 ng/mg, while cyclic users had decreases in mucosal TFV concentrations with 3 day removals, although the trough concentrations remained above 100 ng/mg and quickly returned to concentrations similar to continuous users with IVR re-insertion. Assuming a CVF density of about 1 (1 mg = 0.001 mL), mean TFV concentrations in CVF of TFV/LNG continuous ring users would be around $10^6$ ng/mL, compatible with TFV gel CVF concentrations and prevention of HIV infection [56, 57, 59, 60].

Unlike in the previous study [6], we also measured TFV concentrations in rectal fluid. We found very low concentrations in rectal fluid, which is consistent with past studies of vaginal and rectal administration of TFV gel in macaques [61].

In past studies, tissue concentrations of TFV following the administration of TFV vaginal gel as a single dose, two doses, or 14 daily doses, with or without intercourse, were highly variable ranging from 5.3 ng/mg– 258 ng/mg [56–58]. High TFV-DP concentrations were correlated with TFV concentrations of at least 10 ng/mg in tissue [56–58]. In this study, both continuous and interrupted TFV/LNG IVR users maintained mean and median TFV tissue concentrations above 10 ng/mg. The dispersion of the data observed at end-of-treatment and after final ring removal may be due to differences in the rate of in vivo TFV release associated with vaginal microbial community states, as previously reported [14]. Although the *in vivo* release rates of TFV and LNG were not statistically different between the continuous and cyclic dosing groups (p values 0.65 and 0.22 respectively) (reported in [14]), higher than expected release rates were found associated to community state IV, poly-diverse, vaginal microbiota.

We did not obtain tissue for TFV PK at the end of the 3 day interruption in cyclic users, but found that for all IVR users, TFV tissue concentrations decreased below 10 ng/mg within 48 hours of IVR removal at the end of the study.

The benchmark of 1,000 fmol/mg for TFV-DP levels in tissue comes from PK and efficacy studies of TFV gel in macaques, demonstrating that TFV gel, when applied 30 minutes [62] or even 3 days [63] prior to simian human immunodeficiency virus (SHIV) challenge, protected all or the majority of macaques, respectively. We found no impact of dosing regimen on TFV-DP concentrations. Within 72 hours of IVR insertion, mean and median tissue concentrations of TFV-DP in both the continuous and cyclic dosing cohorts ranged from 2314–4273 fmol/mg. Importantly TFV-DP levels remained high between 2–5 days after IVR removal at the end of treatment (Fig 3B). Thus, our data demonstrate that both dosing regimens for the TFV/LNG IVR resulted in high concentrations of TFV-DP in CV tissues, which remained high even after IVR removal.

The inhibitory activity of the vaginal secretions against HIV-1 at baseline was variable and similar to previous data in healthy women [6, 25, 64, 65]. Consistent with the high vaginal concentrations of TFV, once participants were exposed to TFV containing IVRs, the inhibitory activity of vaginal fluid against HIV-1 *in vitro* increased significantly from baseline at month 1

and at the end of treatment and similar to levels seen with use of TFV vaginal gel [6, 25, 66, 67]. Consistent with the PK data, we did not find any significant impact on HIV replication *in vitro* with rectal fluids obtained in the presence of TFV/LNG IVR use.

Our *ex vivo* CV tissue infection modeling surrogate showed that with TFV exposure, HIV p24 antigen production decreased by greater than 10 fold compared to baseline or placebo treated tissues. This did not reach statistical significance possibly due to the high inter and intra assay variability reported in p24 antigen production in *ex vivo* biopsy challenge experiments, especially at baseline and the small sample size [68]. However, we found a statistically significant correlation between inhibition levels and TFV concentrations in tissue.

Although not the primary outcomes of the trials, topical TFV gel reduced acquisition of genital HSV-2 compared to placebo in two phase IIb trials [69–71]. In this study, as in our first study of the TFV/LNG IVR [6], levels of TFV in the CV aspirate were consistent with HSV-2 inhibition *in vitro* and *in vivo* [70, 72], but were not sufficient to consistently inhibit HSV-2 in the Vero cell plaque reduction assay, as dilution of the CVF with 400 μL was required to avoid cytotoxicity and perform the assay [67].

Our *post hoc* analysis using a HEC1a cell line assay and HSV-2 DNA PCR as an endpoint was able to demonstrate a significant and specific inhibition both using continuous and categorical endpoint variables. Furthermore, we found a significant PK/PD correlation between TFV concentrations in vaginal fluid and *in vitro* inhibition of HSV-2.

Similar to previous findings after approximately 2 weeks of use, using the TFV/LNG IVR for its intended duration of approximately 90 days showed no safety concerns, achieved LNG and TFV PK benchmarks, displayed PD surrogates of anti-HIV-1, HSV-2 and contraceptive activity, and demonstrated high TFV-DP tissue levels supporting potential pharmacological forgiveness. Importantly, we also showed that removing the IVR cyclically seemed to offer no advantage to regulating breakthrough bleeding, which in any case occurred in only few women due to the use of a micro-dose of LNG.

## Conclusions

The TFV/LNG IVR was safe and well tolerated, and met PK/PD pre-specified levels, compatible with HIV, HSV and contraceptive activity. This IVR is capable of releasing in a controlled manner two very different drugs in amounts that differ by roughly 500-fold. Drug release from the IVRs achieved high local concentrations and low plasma levels, minimizing potential systemic AEs. Maintaining high local concentrations post IVR removal (dosing forgiveness) and multipurpose prevention of STIs and pregnancy are value-added features of this unique IVR. Because cyclic dosing did not offer an advantage in the menstrual bleeding pattern over continuous use, our data currently would support a preferred recommendation of continuous IVR use, as this regimen shows a more consistent PK/PD pattern. Long-lasting protective levels of TFV-DP in tissue, however, would also support ring removal for short periods. The ultimate efficacy of this unique MPT ring must be tested in future prevention trials.

## Supporting information

**S1 Checklist. CONSORT 2010 checklist of information to include when reporting a randomised trial**\*.
(DOC)

**S1 Table. Schedule of evaluations.**
(DOCX)

**S2 Table. CD4 and CCR5 positive cell density in cryopreserved samples.**
(DOCX)

**S3 Table. Pharmacokinetic parameters.**
(DOCX)

**S4 Table. Comparison of paired changes from baseline in percent HSV-2 inhibition by vaginal and rectal fluids using vero cell assay.**
(DOCX)

**S1 File. Supplemental methods.**
(DOCX)

**S1 Fig. CONSORT 2010 flow diagram.**
(DOC)

**S1 Protocol.**
(DOCX)

## Author Contributions

**Conceptualization:** Andrea R. Thurman, Vivian Brache, Neelima Chandra, Terry Jacot, Nazita Yousefieh, Meredith R. Clark, Jill L. Schwartz, Elizabeth Tolley, Gustavo F. Doncel.

**Data curation:** Andrea R. Thurman, Leila Cochon, Louise A. Ouattara, Neelima Chandra, Terry Jacot, Nazita Yousefieh, Meredith R. Clark, Homaira Hanif, Jill L. Schwartz, Mark A. Marzinke, David W. Erikson, Urvi Parikh, Betsy C. Herold, Raina N. Fichorova, Elizabeth Tolley, Gustavo F. Doncel.

**Formal analysis:** Andrea R. Thurman, Louise A. Ouattara, Neelima Chandra, Terry Jacot, Nazita Yousefieh, Meredith R. Clark, Melissa Peet, Jill L. Schwartz, Mark A. Marzinke, David W. Erikson, Urvi Parikh, Betsy C. Herold, Raina N. Fichorova, Elizabeth Tolley, Gustavo F. Doncel.

**Funding acquisition:** Andrea R. Thurman, Meredith R. Clark, Jill L. Schwartz, Gustavo F. Doncel.

**Investigation:** Andrea R. Thurman, Vivian Brache, Leila Cochon, Louise A. Ouattara, Neelima Chandra, Terry Jacot, Nazita Yousefieh, Melissa Peet, Mark A. Marzinke, David W. Erikson, Urvi Parikh, Betsy C. Herold, Raina N. Fichorova, Elizabeth Tolley, Gustavo F. Doncel.

**Methodology:** Andrea R. Thurman, Vivian Brache, Leila Cochon, Louise A. Ouattara, Neelima Chandra, Terry Jacot, Nazita Yousefieh, Mark A. Marzinke, David W. Erikson, Urvi Parikh, Betsy C. Herold, Raina N. Fichorova, Elizabeth Tolley.

**Project administration:** Andrea R. Thurman, Meredith R. Clark, Homaira Hanif, Susan Ju, Urvi Parikh, Gustavo F. Doncel.

**Resources:** Vivian Brache, Leila Cochon, Louise A. Ouattara, Neelima Chandra, Terry Jacot, Nazita Yousefieh, Melissa Peet, Susan Ju, Mark A. Marzinke, David W. Erikson.

**Supervision:** Andrea R. Thurman, Meredith R. Clark, Homaira Hanif, Jill L. Schwartz, Susan Ju, Urvi Parikh, Betsy C. Herold, Raina N. Fichorova, Elizabeth Tolley, Gustavo F. Doncel.

**Validation:** Andrea R. Thurman, Louise A. Ouattara, Terry Jacot, Meredith R. Clark, Melissa Peet, Jill L. Schwartz, David W. Erikson, Urvi Parikh, Betsy C. Herold, Raina N. Fichorova, Gustavo F. Doncel.

**Writing – original draft:** Andrea R. Thurman, Louise A. Ouattara, Meredith R. Clark, Raina N. Fichorova, Gustavo F. Doncel.

**Writing – review & editing:** Andrea R. Thurman, Vivian Brache, Leila Cochon, Louise A. Ouattara, Neelima Chandra, Terry Jacot, Nazita Yousefieh, Meredith R. Clark, Melissa Peet, Homaira Hanif, Jill L. Schwartz, Susan Ju, Mark A. Marzinke, David W. Erikson, Urvi Parikh, Betsy C. Herold, Raina N. Fichorova, Elizabeth Tolley, Gustavo F. Doncel.

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
