## [Decision Letter · Decision Letter 0]

22 Mar 2022

PONE-D-21-23955

Randomized, Placebo Controlled Phase I Trial of Safety, Pharmacokinetics, Pharmacodynamics and Acceptability of a 90 day Tenofovir Plus Levonorgestrel Vaginal Ring in Women

PLOS ONE

Dear Dr. Thurman,

Thank you for submitting your manuscript to PLOS ONE. After careful consideration, we feel that it has merit but does not fully meet PLOS ONE’s publication criteria as it currently stands. Therefore, we invite you to submit a revised version of the manuscript that addresses the points raised during the review process.

The manuscript has been evaluated by two reviewers, and their comments are available below.

The reviewers have raised a number of concerns. They feel the writing style and structure of the manuscript should be revised to aid readability and clarity. In additional they request improvements to the reporting of methodological aspects of the study, for example, they request more details about the statistical analyses presented and justification for the use of these analyses.

Could you please carefully revise the manuscript to address all comments raised?

We look forward to receiving your revised manuscript.

Kind regards,

Jamie Royle, PhD

Associate Editor

PLOS ONE

“This study and the clinical development of the TFV/LNG ring were supported by the United States Agency for International Development (USAID) with funds from The U.S. President's Emergency Plan for AIDS Relief (PEPFAR) under Cooperative Agreements (AID-OAA-A-10-00068, AID-OAA-A-14-00010, and AID-OAA-A-14-00011). The Endocrine Technologies Core (ETC) at the Oregon National Primate Research Center (ONPRC) is supported by NIH Grant P51 OD011092 awarded to ONPRC. The contents are the sole responsibility of the authors and do not necessarily reflect the views of their institutions, PEPFAR, USAID or the United States Government.  The funders had no role in study design, data collection and analysis, decision to publish, or preparation of the manuscript.  No author reports any conflict of interest or any competing interests which interfered with the conduct of this study or interpretation of results.”

“This study and the clinical development of the TFV/LNG ring were supported by the United States Agency for International Development (USAID) with funds from The U.S. President's Emergency Plan for AIDS Relief (PEPFAR) under Cooperative Agreements (AID-OAA-A-10-00068, AID-OAA-A-14-00010, and AID-OAA-A-14-00011). The Endocrine Technologies Core (ETC) at the Oregon National Primate Research Center (ONPRC) is supported by NIH Grant P51 OD011092 awarded to ONPRC. The contents are the sole responsibility of the authors and do not necessarily reflect the views of their institutions, PEPFAR, USAID or the United States Government.  The funders had no role in study design, data collection and analysis, decision to publish, or preparation of the manuscript.  No author reports any conflict of interest or any competing interests which interfered with the conduct of this study or interpretation of results.”

Reviewers' comments:

Reviewer's Responses to Questions

**Comments to the Author**

1. Is the manuscript technically sound, and do the data support the conclusions?

Reviewer #1: Yes

Reviewer #2: Yes

2. Has the statistical analysis been performed appropriately and rigorously? 

Reviewer #1: Yes

Reviewer #2: No

3. Have the authors made all data underlying the findings in their manuscript fully available?

Reviewer #1: Yes

Reviewer #2: Yes

4. Is the manuscript presented in an intelligible fashion and written in standard English?

Reviewer #1: Yes

Reviewer #2: No

5. Review Comments to the Author

Reviewer #1: A 4-arm randomized, placebo-controlled phase I safety trial was conducted which reported pharmacokinetics, pharmacodynamics and acceptability. All IVRs were safe and no serious adverse events were encountered.

Minor revisions:

1- Page 10: Indicate if adverse events were collected according to a standardized method, e.g. by CTCAE.

2- Define the abbreviations SD and STD at their first occurrence.

3- Page 25, line 4: State the statistical method from which the paired p-values were estimated.

4- Provide a comprehensive statistical analysis section. State all statistical methods used, including chi square tests, and Spearman correlations.

5- To improve transparency, on each table indicate the statistical testing method(s) used to estimate the p-values.

Reviewer #2: MAJOR COMMENTS

As general comment, I highly recommend a complete re-writing of the manuscript to obtain a more formal, better structured text. As an example, try to keep the same scheme for describing methods in all the sub-sections of this section (as a guidance: which type of sample, when in the study, what to analyze in it). Finally, try to avoid so many abbreviations throughout the text (and always in titles), mainly the non-standard abbreviations (CM, CV, CVF, CVL….)

Statistical methods must be reviewed. The use of parametrical test with low sample size and disbalanced groups must be very well justified…if possible!

Finally, try to follow the style of your previous paper (PLOS one 2018), with some exceptions below recommended.

MINOR COMMENTS

Pg.5, Ln.12-23: The difference between the CONRAD 128 (15 days use) and this study (up to 90 days) is not apparent enough after a quick reading. Then, the wording should be improved. As a suggestion, split it in two paragraphs: The first one to summarize the CONRAD 128, including some results from this study to make more obvious that this study was completed. The second paragraph, dedicated to the current paper as a subsequent study from the former, finalizing with the objectives.

Pg.6. A scheme of activities and short description of visits is mandatory at the beginning of Methods sections. Throughout the section, refer to this figure when necessary and avoid the continuous quoting of “…prior to insertion….”, “….at the end of….”, or similar sentences.

Pg.6,Ln.14. The inclusion/exclusion criteria must be summarized.

Pg.6,Ln.20-22: The 4 arms are explained (and better explained) in Pg.8,Ln.8-11.

Pg.7, Ln.3: It is not clear if the ring was inserted at a specific time of the cycle (e.g., Days 1 to 5 of menstruation) or was inserted at any moment of the cycle, regardless the menstrual status.

Pg.7, Ln.4: Please explain the different extraction regimen “based on the randomization assignment”.

Pg.7, Ln5: Please explain how cervicovaginal (CV) tissue is used for PK purposes. In addition, abbreviation “CV” must be clarified since this is the first time of use (cervicovaginal).

Pg.7, Ln14: Please clarify when did you collect CM ….end of the month 1, 2 or 3 visit…

Pg.9,Ln11: …designed to release approximately 20 u/day. See Ln.6.

Pg.10,Ln 22: As stated above, please: Type of sample > when > what to do.

Pg.11,Ln 3-4: Since no further comments will be included, please consider to delete this sub-section.

Pg.11, Ln 7-13: This is an example of re-witting and simplification, as suggested above: At visits 4 and 29, a CV fluid lavage was collected in 4ml normal saline after speculum insertion and lavage of the cervical fornices and vaginal walls, avoiding spraying directly into the cervical os. The concentration of soluble proteins in the supernatant was then measured by multiplex electrochemiluminescence assay and ELISA, using procedures established under accreditation by the College of American Pathologists [6].

Pg.11,Ln21: ….a subset of XX participants….. Though again, since no further comments will be included, please consider to delete this sub-section.

Pg.12, Ln4: Strictly speaking, “Pharmacokinetics” (PK) is currently defined as the study of the time course of drug absorption, distribution, metabolism, and excretion. The measurement of drug concentration in tissues or the quantification of binding protein cannot be considered as PK. The sub-sections must be re-written following the previous recommendation and the times of quantification must be included on the text (Supplemental is not appropriate for this purpose).

Pg.13,Ln.8: This sub-section is described in much more detail than the other sub-sections. The style must be homogenized throughout the manuscript. Since there is a Supplemental, this sub-section must be reduced in agreement with the previous.

Pg.15,Ln.3: Contraceptive efficacy can only be measured by the follow-up for a long enough time of a cohort of sexually active women (please, see Pearl Index). In this sub-section the PD of LNG is addressed, as well described in its title.

Pg.15, Ln.10: Already described in Pg.7,Ln.21-23. Improve this content in pg.7 and delete this subsection in pg. 15.

Pg.15, Ln.16: Please confirm that the study was authorized by IRB an HA without a formal calculation of sample size.

Pg.16,Ln.1: A real PK analysis must include a description of the concentration/time curve….

Pg.16,Ln.11: “For select PD…” it is not acceptable, must be for all, unless a good explanation of selection criteria is provided.

Pg.16,Ln6-20: The statistical methods must be reviewed. The proposal of parametrical tests sound inadequate for such a low sample size and disbalanced between groups.

Pg.16.Ln.12: The ANCOVA model must be appropriately described. But prior, review the appropriateness of this statistical method.

Pg.16.Ln.17: It is not clear if paired t-test is used for the comparison of more than 2 times, which is inappropriate.

Pg.16.Ln.21: No description of the safety analysis is missing.

Pg.17,Ln.11: Please clarify: If the subject was allocated in one arm, the subject was randomized and therefore enrolled (by definition). Do you mean that she was not included on the analysis? Was the analysis ITT or PP or both? (Not specified on the Statistitics sub-section).

Pg.17,Ln.13: Demographic characteristics of participants.

Pg.20,Ln.1-2: Please delete, not necessary.

Pg.20,Ln.5-15: Please, use a table to show the exposed population by arm.

Pg.20,Ln.16: Please, include a sentence that all the AEs recorded were considered as TEAE (if so!).

20.19: In accordance with this sentence, the incidence of TEAE was lower in interrupted treatment than in placebo.

20.1-….:Please, use the classification by ICH-E2A to show the safety results, including the classification of AE by severity, causality and seriousness. Additionally, expectedness. And this classification should be explained in Methods.

23.13: Delete “(all…0.05)”

23.15: See 23.13.

23.16: of ….N…. samples. And the same in next pages, when p is > 0.05. The statistical significance level at p<0.05 is already established in Methods/Statistical methods (16.20).

24. To simplify this one and the following tables, you can choose between either Mean&SD (if data follow a Normal distribution) or alternatively, Median&Min-Max (if not normal).

27.1-9: Considering the limited sample size in the placebo group, the use of percentages is not appropriate. The description in absolute values (1/5… One of the 5 …. Etc.) is more adequate and simplifies the paragraph.

27.12-17: This comments correspond to Discussion.

28.10: Considering the high number of tables, avoid Table 5 and include only relevant results in the text.

30.2: Results must be shown. A figure is recommended.

34. This table can be included on the text.

38.19-23. The statistical methods (ROC, Fisher, Spearman…) must be described only on the Methods section, not in Results.

41.4. The higher frequency of TEAE in interrupted than in continuous treatment (Pg.21,Ln.6) must be discussed at any point of this section.

41.10. The inclusion of a not sexually active participant should be justified or commented.

42.2. How was resolved? With the removal of the ring? With additional treatment?

42.23. Avoid the direct relationship of bleeding irregularities and anemia. Menstrual irregularities can involve also amenorrhea, or irregular bleeding not necessarily excessive and hence not related with anemia. From other perspective, the potential reason for discontinuation is the irregular bleeding pattern, not the abnormal laboratory finding.

44.3. Clarify that the plasma levels refers to the steady state. Not necessary to specify the method of quantification.

44.8. The use of the same method of quantification does not guarantee that independent results can be directly compared.

Fig.2-3 The use of concentration/time CURVES is recommended (instead of bars). Please, taking in consideration that X-axis is time.

6. PLOS authors have the option to publish the peer review history of their article (what does this mean?). If published, this will include your full peer review and any attached files.

Reviewer #1: No

Reviewer #2: No

---

## [Author Response · Author response to Decision Letter 0]

22 Apr 2022

Andrea Thurman, MD

Professor of Obstetrics and Gynecology

Eastern Virginia Medical School/CONRAD

601 Colley Ave, Norfolk, VA 23507

Email: thurmaar@evms.edu, Phone: 210-380-5241

19 APR 2022

Dear Dr. Royle, Editorial Team Members and Reviewers,

Thank you for your thoughtful review of our manuscript, PONE-D-21-23955, “Randomized, Placebo Controlled Phase I Trial of Safety, Pharmacokinetics, Pharmacodynamics and Acceptability of a 90 day Tenofovir Plus Levonorgestrel Vaginal Ring in Women”. We have addressed all of the reviewer’s comments and recommended edits below. We point to the page and line number(s) of the changes in a revised, tracked version of the manuscript. We hope that these changes make our manuscript acceptable for publication.

Our amended funding statement is as follows: This study and the clinical development of the TFV/LNG ring were supported by the United States Agency for International Development (USAID) with funds from The U.S. President's Emergency Plan for AIDS Relief (PEPFAR) under Cooperative Agreements (AID-OAA-A-10-00068, AID-OAA-A-14-00010, and AID-OAA-A-14-00011). The Endocrine Technologies Core (ETC) at the Oregon National Primate Research Center (ONPRC) is supported by NIH Grant P51 OD011092 awarded to ONPRC. Tenofovir active pharmaceutical ingredient was donated by Gilead Sciences (Foster City, CA, USA).The contents are the sole responsibility of the authors and do not necessarily reflect the views of their institutions, PEPFAR, USAID or the United States Government. The funders had no role in study design, data collection and analysis, decision to publish, or preparation of the manuscript. No author reports any conflict of interest or any competing interests which interfered with the conduct of this study or interpretation of results.

RESPONSE: We confirm that the manuscript is formatted per PLOS One’s style requirements. Headings are 18 font, 2nd level headings are 16 font, 3rd level headings are 14 font. Text is double-spaced at 12 font. The references are cited using Plos style in EndNote. Figures and Tables and Supplemental materials are labeled as per the journal style format.

“This study and the clinical development of the TFV/LNG ring were supported by the United States Agency for International Development (USAID) with funds from The U.S. President's Emergency Plan for AIDS Relief (PEPFAR) under Cooperative Agreements (AID-OAA-A-10-00068, AID-OAA-A-14-00010, and AID-OAA-A-14-00011). The Endocrine Technologies Core (ETC) at the Oregon National Primate Research Center (ONPRC) is supported by NIH Grant P51 OD011092 awarded to ONPRC. The contents are the sole responsibility of the authors and do not necessarily reflect the views of their institutions, PEPFAR, USAID or the United States Government. The funders had no role in study design, data collection and analysis, decision to publish, or preparation of the manuscript. No author reports any conflict of interest or any competing interests which interfered with the conduct of this study or interpretation of results.”

RESPONSE: We have amended funding information and placed it under the Funding Statement section.

“This study and the clinical development of the TFV/LNG ring were supported by the United States Agency for International Development (USAID) with funds from The U.S. President's Emergency Plan for AIDS Relief (PEPFAR) under Cooperative Agreements (AID-OAA-A-10-00068, AID-OAA-A-14-00010, and AID-OAA-A-14-00011). The Endocrine Technologies Core (ETC) at the Oregon National Primate Research Center (ONPRC) is supported by NIH Grant P51 OD011092 awarded to ONPRC. The contents are the sole responsibility of the authors and do not necessarily reflect the views of their institutions, PEPFAR, USAID or the United States Government. The funders had no role in study design, data collection and analysis, decision to publish, or preparation of the manuscript. No author reports any conflict of interest or any competing interests which interfered with the conduct of this study or interpretation of results.”

RESPONSE: There are no funding related statements in the text of the manuscript. All funding related statements are now in the funding section. 

RESPONSE: This amended text is included in the body of the cover letter. 

RESPONSE: There are no ethical or legal restrictions on de-identified study endpoint data. The rest is not applicable.

RESPONSE: We will deposit de-identified data with PLOS One as part of the manuscript submission.

Reviewers' comments:

Reviewer's Responses to Questions

Comments to the Author

1. Is the manuscript technically sound, and do the data support the conclusions?

Reviewer #1: Yes

Reviewer #2: Yes

RESPONSE: Thank you

2. Has the statistical analysis been performed appropriately and rigorously? 

Reviewer #1: Yes

Reviewer #2: No

RESPONSE: We made the edits from reviewer #2 to the statistical methods below

3. Have the authors made all data underlying the findings in their manuscript fully available?

Reviewer #1: Yes

Reviewer #2: Yes

RESPONSE: Thank you – we will also deposit the anonymized data as per requested above by the journal requirements. 

 4. Is the manuscript presented in an intelligible fashion and written in standard English?

Reviewer #1: Yes

Reviewer #2: No

RESPONSE: We made the edits and clarifications requested by reviewer #2 below.

5. Review Comments to the Author

Reviewer #1: A 4-arm randomized, placebo-controlled phase I safety trial was conducted which reported pharmacokinetics, pharmacodynamics and acceptability. All IVRs were safe and no serious adverse events were encountered.

Minor revisions:

1- Page 10: Indicate if adverse events were collected according to a standardized method, e.g. by CTCAE.

RESPONSE: We clarified on page 12 of the revised tracked manuscript that per protocol, adverse events were graded for severity, using the DAIDS tables for grading the severity of adverse events (http://rsc.tech-res.com/clinical-research-sites/safety-reporting/daids-grading-tables) and relationship to study product or study procedures (reported as related versus not related). Adverse events were coded with the appropriate MDR code for the clinical study report and for data presentation. 

2- Define the abbreviations SD and STD at their first occurrence.

RESPONSE: “Standard deviation” is defined at the first occurrence, Table 1, and is uniformly abbreviated as SD now in Tables 1, 3, 4, 6, 7 and 8.

3- Page 25, line 4: State the statistical method from which the paired p-values were estimated.

RESPONSE: We added in the Sample size and statistical analyses section (pages 18 – 20) that the Wilcoxon signed rank sum test was used to compare paired changes from baseline, pre-insertion safety endpoints to the end of treatment values (e.g. density of tissue lymphocytes and concentrations of secreted soluble proteins) as these data are not normally distributed. Similarly, the Kruskall-Wallis test was used to compare safety endpoints among the 4 independent treatment/dosing groups at the end of treatment. We put this clarification in the Methods section because it applies to the statistical analyses presented for both immune cells and cytokines.

4- Provide a comprehensive statistical analysis section. State all statistical methods used, including chi square tests, and Spearman correlations.

RESPONSE: We have added on pages 19 and 20 all statistical tests used to compare safety, PK and PD data. We also added the correlation methods used to correlate PK and PD variables.

5- To improve transparency, on each table indicate the statistical testing method(s) used to estimate the p-values.

RESPONSE: The statistical test for calculating the p values has been added to the Table legend for Tables 2, 3, 4, 5, and 7. Additional information about the statistical analyses has been added to the methods section, pages 19 and 20 in the revised tracked manuscript.

Reviewer #2: MAJOR COMMENTS

As general comment, I highly recommend a complete re-writing of the manuscript to obtain a more formal, better-structured text. As an example, try to keep the same scheme for describing methods in all the sub-sections of this section (as a guidance: which type of sample, when in the study, what to analyze in it). Finally, try to avoid so many abbreviations throughout the text (and always in titles), mainly the non-standard abbreviations (CM, CV, CVF, CVL….)

RESPONSE: We have edited the manuscript to a more formal writing style. In the past, many reviewers have asked that we either use a first person active language or that we use the third person past tense style. We have removed abbreviations from the section headings. We do believe that CV, CVF and CVL are standard abbreviations for this field and type of Phase I study and would therefore like to keep these abbreviations. In addition they are used frequently throughout the text. We have now spelled out cervical mucus. We have formatted the methods text as requested and agree that presenting sample type, when in the study and then specified in the statistical section the tests used to compare each endpoint. Due to the length of the manuscript, we added additional methods details in a supplemental methods section.

Statistical methods must be reviewed. The use of parametrical test with low sample size and disbalanced groups must be very well justified…if possible!

RESPONSE: As also requested by reviewer #1, we have added additional detail to the statistical methods section, pages 18 – 20, specifying which tests were used for each endpoint comparison. We added on page 19 that all data sets were examined for normality using the PROC UNIVARIATE procedure in SAS and examining the histogram, Q plot and the Shapiro Wilk statistic. All continuous variables were not normally distributed and therefore appropriate non-parametric statistical testing was performed. We also note on page 19 that the sample size for this phase 1 study, like most phase 1 studies and particularly those done previously in this field or specifically on this MPT product, was primarily based on the size of similar studies MPT IVRs [1]. For statistical comparisons, we note on page 19, lines 7 – 11, that the primary objective of the study, safety, was compared using exact statistical methods whenever possible for the reporting of TEAEs between treatment groups. For transparency, we have also added the Fisher exact test p values for proportion of participants reporting various TEAE categories to Table 2. We acknowledged on page 19, lines 9 – 11, that other statistical comparisons must be interpreted with caution for this phase 1 study due to the limited sample size, particularly for exploratory modeled PD objectives. 

Finally, try to follow the style of your previous paper (PLOS one 2018), with some exceptions below recommended.

RESPONSE: As requested we have changed the methods section of this manuscript to reflect the style of our previous PLOS One ring manuscript (2018), explaining at what visits samples were collected, the sample collection specifics and then the sample analysis details. Because this is extensive information, we have also included a supplemental methods section to reduce main manuscript length. 

MINOR COMMENTS

Pg.5, Ln.12-23: The difference between the CONRAD 128 (15 days use) and this study (up to 90 days) is not apparent enough after a quick reading. Then, the wording should be improved. As a suggestion, split it in two paragraphs: The first one to summarize the CONRAD 128, including some results from this study to make more obvious that this study was completed. The second paragraph, dedicated to the current paper as a subsequent study from the former, finalizing with the objectives.

RESPONSE: We agree that this is a very important distinction to make between the initial first-in-woman study of 15 days of continuous use (CONRAD 128) [1] and this first study of the TFV/LNG IVR for the full 90 day duration. We have added these details, as suggested in two paragraphs, to the introduction on pages 5 and 6 of the revised tracked manuscript.

Pg.6. A scheme of activities and short description of visits is mandatory at the beginning of Methods sections. Throughout the section, refer to this figure when necessary and avoid the continuous quoting of “…prior to insertion….”, “….at the end of….”, or similar sentences.

RESPONSE: We have moved the sentence “The study visits and procedures are summarized in Supplementary Table 1” to the beginning of the methods section, clinical study section, page 7 lines 1 – 2. Given that this study had 32 visits, we believe the best method to orient the reader to the multiple visits and endpoints is a standard schedule of evaluations table, with which most clinical study researchers are familiar. Throughout the manuscript, when we refer to visits, we have now cited Supplemental Table 1. Plos One format style has Table 1 describe the study population and therefore we chose, as we have done in our previous Plos One manuscripts, to outline the study visits in Supplemental Table 1. 

Pg.6,Ln.14. The inclusion/exclusion criteria must be summarized.

RESPONSE: We have added additional inclusion and exclusion criteria to the main I/E criteria noted in the methods section, page 7.

Pg.6,Ln.20-22: The 4 arms are explained (and better explained) in Pg.8,Ln.8-11.

RESPONSE: We copied the randomization text on page 8 to better explain the study groups (now on pages 7 and 8) in the overall methods section. This allowed us to delete some of the redundant detail in the randomization section on page 10 of the revised tracked manuscript. 

Pg.7, Ln.3: It is not clear if the ring was inserted at a specific time of the cycle (e.g., Days 1 to 5 of menstruation) or was inserted at any moment of the cycle, regardless the menstrual status.

RESPONSE: We have added in the methods section that IVR insertion at visit 4 was performed in the menstrual/follicular phase on day 6 ± 1 day of the menstrual cycle on page 8, lines 6 – 9. 

Pg.7, Ln.4: Please explain the different extraction regimen “based on the randomization assignment”.

RESPONSE: We clarified, on page 8, lines 10 – 14, that the PK sampling time at visit 5 was based on a randomization where participants had blood and CV tissue for PK obtained at 24 OR 48 OR 72 hours after IVR insertion. We also clarified that this randomization was done because participants could not have CV tissue obtained at all three time points for safety reasons (page 8, lines 12 – 14). When CV tissue biopsies are taken, we place Monsel’s solution on the biopsy site to prevent bleeding. We don’t want the participants to have anything in the vagina for 5 days after a biopsy, including doing a repeat speculum exam to obtain more biopsies. This could result in bleeding from the biopsy site.

Pg.7, Ln5: Please explain how cervicovaginal (CV) tissue is used for PK purposes. In addition, abbreviation “CV” must be clarified since this is the first time of use (cervicovaginal).

RESPONSE: We now define CV at the first presentation on page 8. We also clarified that the tissue biopsies for PK were two vaginal tissue biopsies (page 15, line 7). In the methods section, we detail how the Marzinke central PK lab at Johns Hopkins processed the vaginal tissue for both tenofovir and tenofovir-diphosphate concentrations as part of the pharmacokinetic assessment of the rings. PK parameters are now presented in supplemental Table 3. We also detail in the discussion section (pages 53 – 55) how tissue concentrations of TFV in our current study compare to past studies in women using TFV topical vaginal gel [2-4]. 

Pg.7, Ln14: Please clarify when did you collect CM ….end of the month 1, 2 or 3 visit…

RESPONSE: We have added to the description of the timing of cervical mucus checks, in the methods section on pages 8 – 9 that “To detect ovulation and time cervical mucus assessments, participants returned approximately twice weekly after IVR initiation, to have serum estradiol (E2) and P4 measured. We performed a cervical mucus check for Insler score [7] and sperm penetration assay (modified slide test) [8-12] at the next regularly scheduled visit, usually within 48 hours, if the serum E2 was between 75 – 150 pg/mL. If the serum E2 was over 150 pg/mL, indicating imminent follicular development, we brought the participant in for a cervical mucus check within 24 hours. If the serum E2 did not reach 75 pg/mL before the end of each month (i.e. days 28, 59 and 90), we collected cervical mucus at the end of the month 1 (V13), month 2 (V22) or month 3 (V31) visits (S1 Table)”. 

Pg.9,Ln11: …designed to release approximately 20 u/day. See Ln.6.

RESPONSE: Since the release rate was defined on page 11, line 11 as (20 ug/day), we deleted the redundant phrase on page 11, line 16. 

Pg.10,Ln 22: As stated above, please: Type of sample > when > what to do.

RESPONSE: We have added a standard structure to the methods section, which explains type of sample, when collected and what methods were used to process the samples. We also added reference to the Supplemental Table 1, to direct the reader to the schedule of evaluations.

Pg.11,Ln 3-4: Since no further comments will be included, please consider to delete this sub-section.

RESPONSE: We agree and have deleted this sub-section on page 13 of the revised tracked manuscript.

Pg.11, Ln 7-13: This is an example of re-witting and simplification, as suggested above: At visits 4 and 29, a CV fluid lavage was collected in 4ml normal saline after speculum insertion and lavage of the cervical fornices and vaginal walls, avoiding spraying directly into the cervical os. The concentration of soluble proteins in the supernatant was then measured by multiplex electrochemiluminescence assay and ELISA, using procedures established under accreditation by the College of American Pathologists [6].

RESPONSE: Thank you for the alternative text. We have inserted this text on page 14, lines 3 - 7 based on your recommendation.

Pg.11,Ln21: ….a subset of XX participants….. Though again, since no further comments will be included, please consider to delete this sub-section.

RESPONSE: We agree and have deleted the text on page 14 regarding the individual qualitative interviews as these data will be reported separately.

Pg.12, Ln4: Strictly speaking, “Pharmacokinetics” (PK) is currently defined as the study of the time course of drug absorption, distribution, metabolism, and excretion. The measurement of drug concentration in tissues or the quantification of binding protein cannot be considered as PK. The sub-sections must be re-written following the previous recommendation and the times of quantification must be included on the text (Supplemental is not appropriate for this purpose).

RESPONSE: We agree with the Reviewer that we have not done a full ADME study with the rings, but we have collected enough drug concentration data in different compartments to present some key PK parameters, which illustrate how the rings perform and may achieve the desired effects. We have edited the methods section accordingly and added supplemental Table 3 with PK parameters. Because multiple samples were obtained from multiple matrices, we refer the reader to the S1 Table for each time point, to reduce word length. In addition, to make sure that the PK methods are defined in detail, while respecting the journal’s word limits, we have added ample information to the supplemental methods section on the calculation of PK parameters.

Pg.13,Ln.8: This sub-section is described in much more detail than the other sub-sections. The style must be homogenized throughout the manuscript. Since there is a Supplemental, this sub-section must be reduced in agreement with the previous.

RESPONSE: To assess anti-HSV2 activity (now pages 16 and 17 of the revised tracked manuscript), two methods were used. The Vero cell method has been previously reported ,and therefore we summarized it and added references (page 16). The method using HEC1A epithelial cells is a new method, developed and refined by our group and not previously reported; therefore, we chose to describe this method more comprehensively to enable others to reproduce.

Pg.15,Ln.3: Contraceptive efficacy can only be measured by the follow-up for a long enough time of a cohort of sexually active women (please, see Pearl Index). In this sub-section the PD of LNG is addressed, as well described in its title.

RESPONSE: We agree and have edited this section header on page 18 to clarify that this is LNG PD assessments as surrogates of contraceptive effect. In line with the type of study (Phase I), we have assessed LNG PD effects (e.g. anovulation, thickened cervical mucus) as surrogates of potential contraceptive activity. 

Pg.15, Ln.10: Already described in Pg.7,Ln.21-23. Improve this content in pg.7 and delete this subsection in pg. 15.

RESPONSE: We agree with the recommendation and have added detail and referenced our methods for the assessment of objective adherence biomarkers on page 9 as part of the clinical study methods. This allowed us to delete the redundant text on page 18 of the tracked manuscript.

Pg.15, Ln.16: Please confirm that the study was authorized by IRB and without a formal calculation of sample size.

RESPONSE: We confirm the text in the methods section, page 7 “The study was approved by the Advarra Institutional Review Board (IRB) (#Pro00022358) and Comisiòn Nacional de Bioetica (#030-2017), respectively, and registered with ClinicalTrials.gov (#NCT03279120)”. The statistical summary in the approved protocol (also submitted to Plos One) states in the study synopsis and the statistical analyses summary sections the same verbage as reported in the manuscript, on page 19. Our group does several first in woman and phase I studies and it is the norm for these type of studies to be primarily descriptive in nature. We also note, and have added this to the manuscript, page 19, that the primary objective safety, was measured using TEAEs. For this endpoint, we used the Fisher exact test for comparisons, which is not dependent on sample size. 

Pg.16,Ln.1: A real PK analysis must include a description of the concentration/time curve….

RESPONSE: We agree and have added the calculation of standard PK parameters to the supplemental methods section. We have also included the calculated PK parameter data for TFV in vaginal fluid and vaginal tissue, TFV-DP in vaginal tissue, TFV in plasma and LNG and SHBG in serum as Supplemental Table 3. Furthermore, following the Reviewer’s suggestion below, we have changed the figure to plot the data as concentration/time curves. The PK parameters are compared to established benchmarks for LNG contraceptive concentrations [13] and vaginal fluid TFV [14, 15] and vaginal tissue TFV-DP concentrations in regard to HIV antiviral activity [5, 6], which were previously established from either human cross sectional studies (LNG contraceptive PK [13] and vaginal fluid TFV [14, 15]), or SHIV infection experiments in non-human primates days [5, 6]. These references and the discussion of all these PK benchmarks are included in the discussion section pages 52 - 55. Because a Supplemental Table 3 was added, this makes the rectal and vaginal anti-HSV2 data now Supplemental Table 4.

Pg.16,Ln.11: “For select PD…” it is not acceptable, must be for all, unless a good explanation of selection criteria is provided.

RESPONSE: We apologize for this error. We have specified that for comparison of the anti-HIV inhibition and p24 antigen product between the 4 treatment/dosing groups an ANOVA model was used, page 20.

Pg.16,Ln6-20: The statistical methods must be reviewed. The proposal of parametrical tests sound inadequate for such a low sample size and disbalanced between groups.

RESPONSE: No parametric test has been used to statistically compare data sets following non-Gaussian distributions. As pointed out above, we have added the SAS code (PROC univariate) that we used to determine the normality of each endpoint data set on page 19. We specifically note on page 19 that all continuous variables were not normally distributed and therefore appropriate non-parametric statistical testing was performed. We now also note the statistical test used to compare data as footnotes in all tables. We note in the statistical methods section that the primary objective of the study, safety, was compared using exact statistical methods for the reporting of TEAEs between treatment groups on page 19. We acknowledge in the statistical analyses section that other statistical comparisons must be interpreted with caution for this phase 1 study due to the limited sample size, particularly for exploratory modeled PD objectives.

Pg.16.Ln.12: The ANCOVA model must be appropriately described. But prior, review the appropriateness of this statistical method.

RESPONSE: You are correct that the ANCOVA model, which we have used in the past to categorize “infected” versus “not infected” categories of the p24 antigen production PD model was not used or reported in this manuscript and therefore we have removed this model from the statistical methods section on page 20.

Pg.16.Ln.17: It is not clear if paired t-test is used for the comparison of more than 2 times, which is inappropriate.

RESPONSE: We have clarified in Table 6 that paired changes from baseline to month 1 and paired changes from baseline to month 3 were calculated using the non-parametric Wilcoxon signed rank sum test. We recognize that multiple comparisons could have also been performed, but with only two time points from baseline, paired non-parametric analyses are also accurate. We added additional text in the methods section to clarify this statistical process (page 20, lines 1 – 7) for the HIV inhibition of vaginal fluids endpoint. 

Pg.16.Ln.21: No description of the safety analysis is missing.

RESPONSE: As is typical for phase 1 studies, we clarified in the methods section (Safety Assessments sub-section) that the primary safety measure was treatment emergent adverse events (TEAEs). We added, based on reviewer #1’s comments how TEAEs were graded (NIH/NIAID/DAIDS criteria) and how relationship to study product or study procedures was determined on page 12, lines 8 - 10. The other sub-clinical safety assessments outlined in the methods section include the assessment of the density and phenotype of immune cells in ectocervical tissue and changes from baseline in the concentration of secreted soluble proteins from the CV mucosa.

Pg.17,Ln.11: Please clarify: If the subject was allocated in one arm, the subject was randomized and therefore enrolled (by definition). Do you mean that she was not included on the analysis? Was the analysis ITT or PP or both? (Not specified on the Statistitics sub-section).

RESPONSE: We have added a description of the randomized, treated and evaluable population to the statistical analysis section on page 19, lines 12 - 18 

Pg.17,Ln.13: Demographic characteristics of participants.

RESPONSE: As per Plos One journal style, the title of the figure (in this case Figure 1) is listed after the paragraph in which the figure is referenced, on page 22, line 8. The title of Figure 1 is Disposition of participants. The title of Table 1 is Demographic and baseline characteristics treated population. Per Plos One formatting requirements, titles of tables are listed below the table on page 24.

Pg.20,Ln.1-2: Please delete, not necessary.

RESPONSE: We have deleted this text on page 25, lines 1 and 2.

Pg.20,Ln.5-15: Please, use a table to show the exposed population by arm.

RESPONSE: We agree that this is a more straightforward way to display exposure data and have included this text as data in the main safety table, Table 2.

Pg.20,Ln.16: Please, include a sentence that all the AEs recorded were considered as TEAE (if so!).

RESPONSE: We have included on page 12, lines 6 and 7 of the methods section, page 11, that TEAEs were monitored at every visit, starting with the enrollment visit and therefore all adverse events were considered to be TEAEs, not medical history.

20.19: In accordance with this sentence, the incidence of TEAE was lower in interrupted treatment than in placebo.

RESPONSE: We have clarified, using the Fisher exact statistical comparison of the proportion of participants in each of the 4 treatment/dosing groups reporting various levels of TEAEs and the p values are now displayed in Table 2. This clarifies that while there were trends, although none were statistically significant (page 25, lines 18 – 20).

20.1-….:Please, use the classification by ICH-E2A to show the safety results, including the classification of AE by severity, causality and seriousness. Additionally, expectedness. And this classification should be explained in Methods.

RESPONSE: We have outlined additional detail on the classification of severity of TEAEs (mild, moderate, severe, potentially life threatening and death) and gave the link for the DAIDS tables (http://rsc.tech-res.com/clinical-research-sites/safety-reporting/daids-grading-tables) which were used to grade the severity of TEAEs in this study on page 12, lines 6 - 20. We added definitions used for classifying a TEAE as suspected versus unexpected and serious (page 12, lines 12 – 20). We have added information to the results section that there were no SAEs, no unexpected TEAEs or deaths (page 25, line 16).

Page 23.13: Delete “(all…0.05)”

RESPONSE: This text has been deleted on page 30, line 5.

23.15: See 23.13.

RESPONSE: The text of all p values > 0.07 has been deleted on page 30, line 7.

23.16: of ….N…. samples. And the same in next pages, when p is > 0.05. The statistical significance level at p<0.05 is already established in Methods/Statistical methods (16.20).

RESPONSE: We added the detail that there were 12 ectocervical biopsies which could be split for standard formalin fixation versus cryopreservation (done at the EVMS site) for this analyses (page 30, line 8). We also changed the text to say that there were no statistically significant differences between the CD4 and CCR5 lymphocyte densities between TFV/LNG and placebo IVR groups (page 30, lines 8 and 9). However these conclusions, as we noted, are limited by the small sample size (displayed in Supplemental Table 2).

Page 24. To simplify this one and the following tables, you can choose between either Mean&SD (if data follow a Normal distribution) or alternatively, Median&Min-Max (if not normal).

RESPONSE: We agree that normally distributed data is displayed by Mean and SD and non-parametric data is best displayed by median with either IQR or min max. However, we would like to include both mean/SD and median to give the reader a better understanding of the data.

Page 27.1-9: Considering the limited sample size in the placebo group, the use of percentages is not appropriate. The description in absolute values (1/5… One of the 5 …. Etc.) is more adequate and simplifies the paragraph.

RESPONSE: We clarified in the methods section, that the proportions noted for adherence are the percent of expected days of use (page 9, lines 10 – 17). The percentages are not the percent of participants with perfect use. Specifically, based on self-report and observations of IVR use in the clinic, for the continuous arms, the total number of days with the IVR was determined as (the last day of IVR removal – the first day of IVR insertion + 1) – the total number of days of unintentional IVR removal. The expected number of days with the IVR for continuous use was 90 days. For the interrupted arms, the total number of days with the IVR was determined as (the last day of IVR removal – the first day of IVR insertion +1) – (the total number of days of scheduled and unintentional IVR removal). The expected number of days with the IVR for interrupted use was 84 days. We have added verbage to the results section (page 34, lines 2 and 3 and lines 7 and 8) to clarify that this is the mean proportion of days the IVR was used of the expected days of use. We also categorized the proportion of expected days used as 100%, ≥ 80% to < 100%, and < 80%.

Page 27.12-17: This comments correspond to Discussion.

RESPONSE: We agree and took out the comments from the results section (objective biomarkers) (page 34, lines 13 – 19) and placed these in to the discussion section (page 51, lines 14 – 23) regarding how these objective biomarker data compare to our previous validation study study [16]. 

28.10: Considering the high number of tables, avoid Table 5 and include only relevant results in the text.

RESPONSE: Given that bleeding abnormalities are the main reason that women discontinue effective progestin only based contraceptives [17], we would like to keep Table 5. It contains data that the field is looking for. We proposed a microdose of topical LNG (20ug/day) to reduce menstrual irregularities with this LNG MPT [18], and therefore, data on menstrual cycle impact, reported in this cohort over 3 months of use, represent unique and valuable information. 

30.2: Results must be shown. A figure is recommended.

RESPONSE: We agree and have added the TFV in plasma PK data to Supplemental Table 3. 

34. This table can be included on the text.

RESPONSE: We agree and have included the plasma TFV PK data in Supplemental Table 3

38.19-23. The statistical methods (ROC, Fisher, Spearman…) must be described only on the Methods section, not in Results.

RESPONSE: We agree and have moved the statistical methods used for the post hoc categorization of the anti-HSV2 data to the statistical methods section (page 21, lines 6 – 13). 

41.4. The higher frequency of TEAE in interrupted than in continuous treatment (Pg.21,Ln.6) must be discussed at any point of this section.

RESPONSE: As stated above, we used the Fisher exact test to compare the proportion of participants in each study treatment (TFVLNG or placebo IVR) and each dosing regimen (continuous or interrupted) as this was the primary safety endpoint and we recognize that exact tests should be used when the sample size or cell sizes are low. Although there were some trends noted, there were no statistically significant differences in reports of various levels of TEAEs by participants randomized to the four cohorts studied. We have included the summary sentence: “The TFV/LNG IVR was safe and well tolerated. There were no significant differences in the proportion of participants randomized to one of the four treatment and dosing groups in their reports of various levels of TEAEs and the relatedness of these TEAEs to the study product or study procedures” as the top line assessment in the discussion section, pages 48 and 49.

41.10. The inclusion of a not sexually active participant should be justified or commented.

RESPONSE: We have added to the introduction that the initial first in woman study, CONRAD 128, did not allow sexual activity to limit exposure of a male partner to an IND product (page 6, lines 1 - 3). After proving safety in the first study, we allowed sexual activity in this 3 month use study because it was not feasible to limit participant’s sexual activity for this long. Furthermore topical tenofovir has been used previously as a topical gel by thousands of sexually active women in phase 2/3 efficacy trials [19-21]. However, as the 138 study was still a phase 1 study to assess safety, PK and model PD in women, we did not require participants to be heterosexually active as no endpoint in this study was related to a male partner, nor required timed intercourse and this study was not assessing sero-conversion.

42.2. How was resolved? With the removal of the ring? With additional treatment?

RESPONSE: We have added additional details on page 49, lines 9 – 10, to this participant’s finding and recovery. We removed her IVR and her epithelial disruption completely resolved within 10 days without additional treatment. 

42.23. Avoid the direct relationship of bleeding irregularities and anemia. Menstrual irregularities can involve also amenorrhea, or irregular bleeding not necessarily excessive and hence not related with anemia. From other perspective, the potential reason for discontinuation is the irregular bleeding pattern, not the abnormal laboratory finding.

RESPONSE: We agree and have removed the verbage referring to serum hemoglobin and ferritin concentrations on page 51, lines 1 and 2.

44.3. Clarify that the plasma levels refers to the steady state. Not necessary to specify the method of quantification.

RESPONSE: That is correct, these are steady state concentrations of serum LNG in previous studies of the LNG implant (Norplant). We have added this clarification on page 52, lines 15 and 17.

44.8. The use of the same method of quantification does not guarantee that independent results can be directly compared.

RESPONSE: We completely agree. We have added additional detail on pages 52 and 53, to emphasize that the past contraceptive benchmarks for levonorgestrel were for systemic delivery (implants) and were established using RIA on page 52. Although we were able to use RIA in our first study, this method has been replaced with LC/MS-MS. We have added an additional reference [22] (reference 27 in the tracked revised manuscript), which also notes that RIA and LC/MS-MS quantification are not directly comparable.

Fig.2-3 The use of concentration/time CURVES is recommended (instead of bars). Please, taking in consideration that X-axis is time.

RESPONSE: We have changed the bar graphs showing mean (SD) in Figures 2 and 3 to concentration/time graphs. We have added that the X axis is time. 

Again, we thank the reviewers for their thoughtful comments. We hope that the edits and clarifications added makes our manuscript acceptable for publication in PLOS One.

Sincerely,

Andrea Thurman

REFERENCES FOR REVIEWER RESPONSE LETTER:

1. Thurman AR, Schwartz JL, Brache V, Clark MR, McCormick T, Chandra N, et al. Randomized, placebo controlled phase I trial of safety, pharmacokinetics, pharmacodynamics and acceptability of tenofovir and tenofovir plus levonorgestrel vaginal rings in women. PLoS One. 2018;13(6):e0199778. Epub 2018/06/29. doi: 10.1371/journal.pone.0199778. PubMed PMID: 29953547; PubMed Central PMCID: PMCPMC6023238.

2. Schwartz JL, Rountree RW, Kashuba ADM, Brache A, Creinin M, Poindexter A, et al. A Multi-Compartment, Single and Multiple Dose Pharmacokinetic Study of the Vaginal Candidate Microbicide 1% Tenofovir Gel. PLoS One. 2011;6(10):e25974.

3. Herold BC, Chen BA, Salata RA, Marzinke MA, Kelly CW, Dezzutti CS, et al. Impact of Sex on the Pharmacokinetics and Pharmacodynamics of 1% Tenofovir Gel. Clin Infect Dis. 2016;62(3):375-82. doi: 10.1093/cid/civ913. PubMed PMID: 26508513; PubMed Central PMCID: PMC4706638.

4. Hendrix CW, Chen BA, Guddera V, Hoesley C, Justman J, Nakabiito C, et al. MTN-001: randomized pharmacokinetic cross-over study comparing tenofovir vaginal gel and oral tablets in vaginal tissue and other compartments. PLoS One. 2013;8(1):e55013. doi: 10.1371/journal.pone.0055013. PubMed PMID: 23383037; PubMed Central PMCID: PMC3559346.

5. Parikh UM, Dobard C, Sharma S, Cong ME, Jia H, Martin A, et al. Complete protection from repeated vaginal simian-human immunodeficiency virus exposures in macaques by a topical gel containing tenofovir alone or with emtricitabine. J Virol. 2009;83(20):10358-65. Epub 2009/08/07. doi: JVI.01073-09 [pii]

10.1128/JVI.01073-09. PubMed PMID: 19656878; PubMed Central PMCID: PMC2753130.

6. Dobard C, Sharma S, Martin A, Pau CP, Holder A, Kuklenyik Z, et al. Durable protection from vaginal simian-human immunodeficiency virus infection in macaques by tenofovir gel and its relationship to drug levels in tissue. J Virol. 2011;86(2):718-25. Epub 2011/11/11. doi: JVI.05842-11 [pii]

10.1128/JVI.05842-11. PubMed PMID: 22072766; PubMed Central PMCID: PMC3255839.

7. Insler V, Melmed H, Eichenbrenner I, Serr DM, Lunenfeld B. The Cervical Score: A Simple Semiquantitative Method for Monitoring of the Menstrual Cycle. International Journal of Gynaecology and Obstetrics. 1972;10(6):223-7.

8. Leader A, Wiseman D, Taylor PJ. The prediction of ovulation: a comparison of the basal body temperature graph, cervical mucus score, and real-time pelvic ultrasonography. Fertil Steril. 1985;43(3):385-8. PubMed PMID: 3884396.

9. Nulsen J, Wheeler C, Ausmanas M, Blasco L. Cervical mucus changes in relationship to urinary luteinizing hormone. Fertil Steril. 1987;48(5):783-6. PubMed PMID: 3311823.

10. Abidogun KA, Ojengbede OA, Fatukasi UI. Prediction and detection of ovulation: an evaluation of the cervical mucus score. Afr J Med Med Sci. 1993;22(1):65-9. PubMed PMID: 7839884.

11. Organization WH. The World Health Organization Laboratory Manual for the Examination of Human Semen and Sperm-Cervical Mucus Interaction. Fourth ed. Cambridge, United Kingdom: Cambridge University Press; 1999.

12. Pandya IJ, Mortimer D, Sawers RS. A standardized approach for evaluating the penetration of human spermatozoa into cervical mucus in vitro. Fertil Steril. 1986;45(3):357-65. PubMed PMID: 3949035.

13. Cherala G, Edelman A, Dorflinger L, Stanczyk FZ. The elusive minimum threshold concentration of levonorgestrel for contraceptive efficacy. Contraception. 2016;94(2):104-8. doi: 10.1016/j.contraception.2016.03.010. PubMed PMID: 27000997.

14. Kashuba AD, Gengiah TN, Werner L, Yang KH, White NR, Karim QA, et al. Genital Tenofovir Concentrations Correlate With Protection Against HIV Infection in the CAPRISA 004 Trial: Importance of Adherence for Microbicide Effectiveness. J Acquir Immune Defic Syndr. 2015;69(3):264-9. doi: 10.1097/QAI.0000000000000607. PubMed PMID: 26181703; PubMed Central PMCID: PMC4505741.

15. Karim SS, Kashuba AD, Werner L, Karim QA. Drug concentrations after topical and oral antiretroviral pre-exposure prophylaxis: implications for HIV prevention in women. Lancet. 2011;378(9787):279-81. doi: 10.1016/S0140-6736(11)60878-7. PubMed PMID: 21763939; PubMed Central PMCID: PMC3652579.

16. Jacot TA, Clark MR, Adedipe OE, Godbout S, Peele AG, Ju S, et al. Development and clinical assessment of new objective adherence markers for four microbicide delivery systems used in HIV prevention studies. Clin Transl Med. 2018;7(1):37. Epub 2018/11/08. doi: 10.1186/s40169-018-0213-6. PubMed PMID: 30402770; PubMed Central PMCID: PMCPMC6219998.

17. Polis CB, Hussain R, Berry A. There might be blood: a scoping review on women's responses to contraceptive-induced menstrual bleeding changes. Reprod Health. 2018;15(1):114. Epub 2018/06/27. doi: 10.1186/s12978-018-0561-0. PubMed PMID: 29940996; PubMed Central PMCID: PMCPMC6020216.

18. Xiao BL, Zhang XL, Feng DD. Pharmacokinetic and pharmacodynamic studies of vaginal rings releasing low-dose levonorgestrel. Contraception. 1985;32(5):455-71. Epub 1985/11/01. PubMed PMID: 3936678.

19. Karim QA, Karim SSA, Frohlich JA, Grobler AC, Baxter C, Mansoor LE, et al. Effectiveness and Safety of Tenofovir Gel, an Antiretroviral Microbicide, for the Prevention of HIV Infection in Women. Science Express. 2010;10.1126(1193748):1.

20. Delany-Moretlwe S, Lombard C, Baron D, Bekker LG, Nkala B, Ahmed K, et al. Tenofovir 1% vaginal gel for prevention of HIV-1 infection in women in South Africa (FACTS-001): a phase 3, randomised, double-blind, placebo-controlled trial. Lancet Infect Dis. 2018;18(11):1241-50. Epub 2018/12/07. doi: 10.1016/S1473-3099(18)30428-6. PubMed PMID: 30507409.

21. Marrazzo JM, Ramjee G, Richardson BA, Gomez K, Mgodi N, Nair G, et al. Tenofovir-based preexposure prophylaxis for HIV infection among African women. N Engl J Med. 2015;372(6):509-18. doi: 10.1056/NEJMoa1402269. PubMed PMID: 25651245; PubMed Central PMCID: PMC4341965.

22. Blue SW, Winchell AJ, Kaucher AV, Lieberman RA, Gilles CT, Pyra MN, et al. Simultaneous quantitation of multiple contraceptive hormones in human serum by LC-MS/MS. Contraception. 2018;97(4):363-9. Epub 2018/02/07. doi: 10.1016/j.contraception.2018.01.015. PubMed PMID: 29407362; PubMed Central PMCID: PMCPMC5840044.

---

## [Decision Letter · Decision Letter 1]

31 May 2022

PONE-D-21-23955R1Randomized, Placebo Controlled Phase I Trial of Safety, Pharmacokinetics, Pharmacodynamics and Acceptability of a 90 day Tenofovir Plus Levonorgestrel Vaginal Ring in WomenPLOS ONE

Dear Dr. Thurman,

Thank you for submitting your manuscript to PLOS ONE. After careful consideration, we feel that it has merit but does not fully meet PLOS ONE’s publication criteria as it currently stands. Therefore, we invite you to submit a revised version of the manuscript that addresses the points raised during the review process.

The manuscript has been evaluated by two reviewers, and their comments are available below.

The reviewers have raised a number of major concerns. They request improvements to the reporting of methodological aspects of the study. Thee reviewers also note concerns about the statistical analyses presented. Could you please carefully revise the manuscript to address all comments raised?

We look forward to receiving your revised manuscript.

Kind regards,

Thomas Phillips, PhD

Staff Editor

PLOS ONE

Journal Requirements:

Reviewers' comments:

Reviewer's Responses to Questions

**Comments to the Author**

1. If the authors have adequately addressed your comments raised in a previous round of review and you feel that this manuscript is now acceptable for publication, you may indicate that here to bypass the “Comments to the Author” section, enter your conflict of interest statement in the “Confidential to Editor” section, and submit your "Accept" recommendation.

Reviewer #1: (No Response)

Reviewer #2: All comments have been addressed

2. Is the manuscript technically sound, and do the data support the conclusions?

Reviewer #1: Yes

Reviewer #2: Yes

3. Has the statistical analysis been performed appropriately and rigorously? 

Reviewer #1: Yes

Reviewer #2: Yes

4. Have the authors made all data underlying the findings in their manuscript fully available?

Reviewer #1: Yes

Reviewer #2: Yes

5. Is the manuscript presented in an intelligible fashion and written in standard English?

Reviewer #1: Yes

Reviewer #2: (No Response)

6. Review Comments to the Author

Reviewer #1: Minor revisions: (Page numbers refer to those in the tracked changes version of revision 1.)

1- Page 19, Line 8 and Page 25, Line 19: Clarify the phrase "exact statistical methods."

2- Page 26, Line 4 and Page 29, Line 9: "Fisher's exact."

3- Tables 3, 4, 6, 7 and supplemental tables: Typically median, first and third quartiles are used to summarize non-normally distributed data.

4- The abbreviation STD appears in the supplemental tables.

Reviewer #2: • Contratulations by the improvement of the manuscript.

• Something that must be more obvious from the first reading is that you are comparing LVG/TNF ring vs placebo under TWO regimens of use (continuous vs cyclic), resulting in a 4-arms study. I’d suggest to include these ideas (2 products, 2 regimens) from the very beginning (title, abstract), etc.

• ABSTRACT: Please, follow the structure of introduction, methodology, results, discussion and conclusion (but avoid the section titles).

• Page 6, Line 6: The study hereby presented (CONRAD 138)…….(to better distinguish both studies).

• 7, 15-17: The quotation of some factors is not necessary: ….(e.g. ….infections).

• 9,3: Data of residual drug must be provided!!!!! If I’m not wrong, there are several comments to residual glycerin as adherence marker, but not residual drug. Data form LNG/TNF IVR must be provided, analyzed between regimens (continuous vs cyclic) and discussed if differences are seen.

7. PLOS authors have the option to publish the peer review history of their article (what does this mean?). If published, this will include your full peer review and any attached files.

Reviewer #1: No

Reviewer #2: No

---

## [Author Response · Author response to Decision Letter 1]

9 Jun 2022

Andrea Thurman, MD

Professor of Obstetrics and Gynecology

Eastern Virginia Medical School/CONRAD

601 Colley Ave, Norfolk, VA 23507

Email: thurmaar@evms.edu, Phone: 210-380-5241

8 JUN 2022

Dear Dr. Phillips, Editorial Team Members and Reviewers,

Thank you for your thoughtful second review of our manuscript, PONE-D-21-23955R1, “Randomized, Placebo Controlled Phase I Trial of Safety, Pharmacokinetics, Pharmacodynamics and Acceptability of a 90 day Tenofovir Plus Levonorgestrel Vaginal Ring in Women”. We have addressed all of the additional reviewer’s comments and recommended edits below. We point to the page and line number(s) of the changes in a revised, tracked version of the manuscript. We hope that these changes make our manuscript acceptable for publication.

Reviewer #1: Minor revisions: (Page numbers refer to those in the tracked changes version of revision 1.)

1- Page 19, Line 8 and Page 25, Line 19: Clarify the phrase "exact statistical methods."

ANSWER: We have clarified that we mean the Fisher’s exact test, which is not impacted by sample size. This clarification is now on page 18, line 6 of the revision #2 tracked manuscript

2- Page 26, Line 4 and Page 29, Line 9: "Fisher's exact."

ANSWER: We have made this correction, on page 8, line 22 of the tracked second revision.

3- Tables 3, 4, 6, 7 and supplemental tables: Typically median, first and third quartiles are used to summarize non-normally distributed data.

ANSWER: We have edited Tables 3, 4, 6 and 7 to show medians, with the 25th – 75th interquartile ranges for these non-normally distributed data.

4- The abbreviation STD appears in the supplemental tables.

ANSWER: We have revised the Tables 3, 4, 6 and 7 to only include median and interquartile ranges and deleted mean and standard deviation for these non-normally distributed data. In Table 1, standard deviation is correctly abbreviated as SD.

Reviewer #2: • Congratulations by the improvement of the manuscript.

• Something that must be more obvious from the first reading is that you are comparing LVG/TNF ring vs placebo under TWO regimens of use (continuous vs cyclic), resulting in a 4-arms study. I’d suggest to include these ideas (2 products, 2 regimens) from the very beginning (title, abstract), etc.

ANSWER: We have added the dosing regimen (continuous versus cyclic) in the title of the manuscript. Even with this addition to the title, the title is less than the 250 character limit (204 characters). We also added the clarification that this is a 4 arm study to the abstract and introduction. 

• ABSTRACT: Please, follow the structure of introduction, methodology, results, discussion and conclusion (but avoid the section titles).

ANSWER: We have added an introductory statement to the abstract so that now the abstract has the introduction, methodology, results, discussion and conclusions highlighted in the correct order. We have stayed under the 300 word abstract limit (299 words) with these changes.

• Page 6, Line 6: The study hereby presented (CONRAD 138)…….(to better distinguish both studies).

ANSWER: We agree and made edits to the introduction (page 6 of the tracked second manuscript revision) to distinguish the first-in-woman 14 day study (CONRAD 128) from the current 90 day full duration of treatment study (CONRAD 138).

• 7, 15-17: The quotation of some factors is not necessary: ….(e.g. ….infections).

ANSWER: We agree that since the STI screenings are listed in Supplementary Table 1 and the study protocol, we have taken out the specific listings of exclusionary infections (active HSV-2, Neisseria gonorrhoeae, Chlamydia trachomatis, Trichomonas vaginalis, HIV-1, and/or Hepatitis B). This deletion is found on page 7, lines 17 and 18.

• 9,3: Data of residual drug must be provided!!!!! If I’m not wrong, there are several comments to residual glycerin as adherence marker, but not residual drug. Data form LNG/TNF IVR must be provided, analyzed between regimens (continuous vs cyclic) and discussed if differences are seen.

ANSWER: Thank you for this question. We found an intriguing relationship between TFV release rate from the rings and the composition of the vaginal microbiota. Higher than expected in vivo release rates were observed in women with community state type IVA/B, polydisperse microbiota. We have reported these data in a separate manuscript [Thurman AR, Ravel J, Gajer P, Marzinke MA, Ouattara LA, Jacot T, Peet MM, Clark MR, Doncel GF. Vaginal Microbiota and Mucosal Pharmacokinetics of Tenofovir in Healthy Women Using a 90-Day Tenofovir/Levonorgestrel Vaginal Ring. Front Cell Infect Microbiol. 2022 Mar 8;12:799501. doi: 10.3389/fcimb.2022.799501. PMID: 35350436; PMCID: PMC8957918.] describing detailed correlations with microbial community states and individual bacteria such as Prevotella bivia, Gardnerella vaginalis and lactobacillus species (now reference 14 in the second revision). This finding has also been observed in a study of the TFV and TFV/LNG IVRs in Kenyan women (Dabee et al., submitted). Conversely, release rates were similar and not statistically different between rings used continuously or cyclically. We have now clarifications about the included a statement and referenced this paper in methods (pages 9 and 10 in the tracked second revision) and in the discussion section (page 52, lines 17 – 22). 

Thank you again for your thoughtful review of our manuscript. We hope that the additional revisions will make the manuscript acceptable for publication in PLoS One.

Sincerely,

Andrea Thurman

---

## [Decision Letter · Decision Letter 2]

8 Jul 2022

PONE-D-21-23955R2Randomized, Placebo Controlled Phase I Trial of the Safety, Pharmacokinetics, Pharmacodynamics and Acceptability of a 90 day Tenofovir Plus Levonorgestrel Vaginal Ring used Continuously or Cyclically in WomenPLOS ONE

Dear Dr. Thurman,

Thank you for submitting your manuscript to PLOS ONE. After careful consideration, we feel that it has merit but does not fully meet PLOS ONE’s publication criteria as it currently stands. Therefore, we invite you to submit a revised version of the manuscript that addresses the points raised during the review process.

Specifically, we require manuscripts must be presented in an intelligible fashion and be written in standard English. The reviewer raised multiple English language errors and has concerns about the way you presented your Abstract and the objectives of the study. Please note that PLOS ONE does not provide copyediting or proofs of accepted manuscripts. We therefore recommend that you carefully review your manuscript and correct any errors at this time.

We look forward to receiving your revised manuscript.

Kind regards,

Jianhong Zhou

Staff Editor

PLOS ONE

Journal Requirements:

Reviewers' comments:

Reviewer's Responses to Questions

**Comments to the Author**

1. If the authors have adequately addressed your comments raised in a previous round of review and you feel that this manuscript is now acceptable for publication, you may indicate that here to bypass the “Comments to the Author” section, enter your conflict of interest statement in the “Confidential to Editor” section, and submit your "Accept" recommendation.

Reviewer #1: All comments have been addressed

Reviewer #2: (No Response)

2. Is the manuscript technically sound, and do the data support the conclusions?

Reviewer #1: (No Response)

Reviewer #2: Yes

3. Has the statistical analysis been performed appropriately and rigorously? 

Reviewer #1: (No Response)

Reviewer #2: Yes

4. Have the authors made all data underlying the findings in their manuscript fully available?

Reviewer #1: (No Response)

Reviewer #2: Yes

5. Is the manuscript presented in an intelligible fashion and written in standard English?

Reviewer #1: (No Response)

Reviewer #2: No

6. Review Comments to the Author

Reviewer #1: (No Response)

Reviewer #2: OVERVIEW

Please adapt the abstract in accordance with below suggested changes.

Avoid abbreviations in titles (including section titles)

Discuss with the Editor the alternative of splitting the manuscript in 2 different but related papers.

MAJOR COMMENTS

The same order of objectives must be followed through the text. I understand that should be the one stated on the title: Safety, PK, PD, Acceptability. Then follow it in all sections: Pg5,Ln 8-9; Methods (Safety-> PK -> PD -> Acceptability), Results, Discussion and Conclusions. Safety must be always the first since it is the primary outcome.

A review of the manuscript by an expert in drug safety / pharmacovigilance is highly recommended. Some of the statements from authors should be carefully reassessed.

MINOR COMMMENTS

Pg1, Ln1: Typo error “59”

Pg4, Ln19-22: The manuscript is long and this paragraph does not add relevant information, please delete.

Pg6,Ln14: Please list exclusion criteria as you did inclusion. When they were evaluated is described elsewhere. Same for the next paragraphs, the visits are already described in S1 Table.

6,9. Randomization is described later.

7.4 Blood extraction timing is part of the PK methods.

7.8 Confirmation of anovulatory effect is part of the PD.

7 …. Etc.

10.5 If “The unit dose for the IVRs…” is true, then the placebo also release TFV and LNG…!!!! To avoid this type of mistake, describe first the similarities (manufacturing place, physical appearance) and then the drug content of the active product.

11.2 In accordance with Good Pharmacovigilance Practice, all AE must be recorded, and subsequently classified. This statement must be included, not only refer to TE (which of course are the relevant).

11. Delete “or …reaction”

12. As above requested, follow the order of sections.

12. 14 CVL abbreviation.

12.22 And CFR compliant also, I hope!!

16.22 Contraception was not investigated, women followed abstinence !!!

17.3-5 This limitation of the study must be transferred to Discussion section.

17.8 Delete (V3…)

17.9 Before this last paragraph, methodology to assess acceptability is missing.

17.19 “No participant…. Is a Safety result, no allocation of population.

31.1 No previous mention to “expulsions” in objectives, nor in methods…. Actually, no mention of expulsion in this paragraph beyond the title….

31.1 If you consider Adherence as part of Acceptability assessment, explain and merge. If you consider as part of follow-up, then must be described before main results: Safety > Pk>Pd>Accept.

45.5 A first paragraph of wrap-up is usually recommended at the beginning of the Discussion.

46.9 Cannot considered out of caution. Taking in consideration the occurrence of genital ulcers in this and in Keller’s studies, this AE must be considered as a serious concern to be carefully evaluated within the present and future clinical development. It would probably be considered as a risk by Health Authorities and probably, additional measures of safety surveillance will be required.

46.12-16 The two potential causes of ulcers are here confounded: 1-drug effect over the epithelium; 2-ring’s mechanical effect. COMMENT: both must be investigated in future trials.

47.8-9 SERIOUS INCONSISTENCY: TEAE means Treatment – Emergent……therefore, by definition, it is considered related with the treatment !!!!!!!!!!!!!!!!!!!!!!!!!!!!!!

7. PLOS authors have the option to publish the peer review history of their article (what does this mean?). If published, this will include your full peer review and any attached files.

Reviewer #1: No

Reviewer #2: **Yes: **J. Algorta

---

## [Author Response · Author response to Decision Letter 2]

11 Jul 2022

Andrea Thurman, MD

Professor of Obstetrics and Gynecology

Eastern Virginia Medical School/CONRAD

601 Colley Ave, Norfolk, VA 23507

Email: thurmaar@evms.edu, Phone: 210-380-5241

11 JUL 2022

Dear Jianhong Zhou, Editorial Team Members and Reviewers,

Thank you for your thoughtful third review of our manuscript, PONE-D-21-23955R2, “Randomized, Placebo Controlled Phase I Trial of Safety, Pharmacokinetics, Pharmacodynamics and Acceptability of a 90 day Tenofovir Plus Levonorgestrel Vaginal Ring in Women”. We have addressed all of the additional reviewer’s comments and recommended edits below. We point to the page and line number(s) of the changes in a revised, tracked version of the manuscript. We hope that these changes make our manuscript acceptable for publication.

This manuscript was originally submitted to PLoS One on 29 JUL 2021. We thank the PLoS One team for continuing to solicit reviewers and academic editors for our work. This appears to be a third set of reviews, with an additional new reviewer and a third academic editor. As expected, there are some contradictions between what previous reviewers wanted added or deleted and the comments of the current reviewers, in particular the new reviewer. We will now delete additions, as recommended by Reviewer 2, which were previously requested in the first and second reviews of this manuscript.

Reviewers' comments:

Reviewer #1: (No Response)

Reviewer #2: OVERVIEW

Please adapt the abstract in accordance with below suggested changes.

Response: Abstract adapted as per the suggestions below

Avoid abbreviations in titles (including section titles)

Response: We have removed all abbreviations from titles and sub-titles. 

Discuss with the Editor the alternative of splitting the manuscript in 2 different but related papers.

Response: Our research group originally submitted this manuscript to PLoS One on July 29, 2021. This manuscript represents the main findings of the CONRAD 138 study. It has gone through multiple academic editors and reviews. In fact, during the year-long PLoS One review, we published a second, related manuscript of the relation between the vaginal microbiota and pharmacokinetics of tenofovir [1]. We very much want to publish the main findings of the CONRAD 138 study and think the way they are presented in the manuscript has conceptual unity. Therefore we prefer not to split the data and manuscript into two related paper. Furthermore, two previous sets of academic editors and reviewers were agreeable to present these data as a single manuscript.

MAJOR COMMENTS

The same order of objectives must be followed through the text. I understand that should be the one stated on the title: Safety, PK, PD, Acceptability. Then follow it in all sections: Pg5,Ln 8-9; Methods (Safety-> PK -> PD -> Acceptability), Results, Discussion and Conclusions. Safety must be always the first since it is the primary outcome.

Response: We have changed the order in the methods and results section to reflect the objectives of the study: Safety, Pharmacokinetics, Pharmacodynamics and Acceptability (effect on the menstrual cycle).

A review of the manuscript by an expert in drug safety / pharmacovigilance is highly recommended. Some of the statements from authors should be carefully reassessed.

Response: The manuscript has been reviewed by Dr. Jill Schwartz, MD, and Dr. Mark Marzinke, PhD, who currently are in charge of pharmacovigilance and pharmacology, respectively, at a large clinical contract research organization and John Hopkins School of Medicine. We have revised terminology regarding TEAE and clarified assessment and recording of all AEs as requested. The study was performed, and therefore reviewed and independently monitored, under an IND (#118,510) from the USFDA.

MINOR COMMMENTS

Pg1, Ln1: Typo error “59”

RESPONSE: The additional “59” in the title was removed

Pg4, Ln19-22: The manuscript is long and this paragraph does not add relevant information, please delete.

RESPONSE: We removed this text. Of note, during the initial review of this manuscript, a previous PLoS One reviewer requested these additional details.

Pg6,Ln14: Please list exclusion criteria as you did inclusion. When they were evaluated is described elsewhere. Same for the next paragraphs, the visits are already described in S1 Table.

RESPONSE: We added the exclusion criteria on page 6, lines 12 - 16. 

Pages 6,9. Randomization is described later.

RESPONSE: We have deleted the details of the randomization explanation from pages 6 and 7, as this is repeated in the randomization section.

7.4 Blood extraction timing is part of the PK methods.

RESPONSE: We have deleted lines 4 – 8 of page 7, as the timing of the PK blood draws after initial IVR insertion is explained in the PK Methods section.

7.8 Confirmation of anovulatory effect is part of the PD.

RESPONSE: We have consolidated the methods we used for assessing levonorgestrel PD in to the Methods section (pages 16 and 17)

10.5 If “The unit dose for the IVRs…” is true, then the placebo also release TFV and LNG…!!!! To avoid this type of mistake, describe first the similarities (manufacturing place, physical appearance) and then the drug content of the active product.

RESPONSE: We have corrected the errors in explaining the active TFV/LNG IVR versus the placebo IVR on page 10, lines 7 – 21. 

11.2 In accordance with Good Pharmacovigilance Practice, all AE must be recorded, and subsequently classified. This statement must be included, not only refer to TE (which of course are the relevant).

RESPONSE: As clarified in the Methods section, page 11, AEs were collected at each visit, starting with the enrollment visit, visit 3. AEs noted during screening (visits 1 and 2) would be medical history AEs. We clarified in the methods section that AEs were collected at each of the 32 visits, starting with visit 3 and were coded with the appropriate MDR code. 

11. Delete “or …reaction”

RESPONSE: We deleted this phrase on page 11, line 18. Of note, this verbage was a requested addition during the first round of reviews. 

12. As above requested, follow the order of sections.

RESPONSE: We have re-ordered the methods section to explain safety (both clinical and sub-clinical safety), PK, PD and acceptability assessments.

12. 14 CVL abbreviation.

RESPONSE: CVL (cervico-vaginal fluid lavage) is now defined on page 13, line 1.

12.22 And CFR compliant also, I hope!!

RESPONSE: Yes, CFR compliance was added, now on page 17, line 16.

16.22 Contraception was not investigated, women followed abstinence !!!

RESPONSE: We agree, and this is why the sub-title is pharmacodynamics surrogate. We have added that potential contraceptive efficacy was modeled by several surrogates to make it clear that we are not directly evaluating contraceptive efficacy or tracking pregnancy.

17.3-5 This limitation of the study must be transferred to Discussion section.

RESPONSE: We have deleted the verbage from the Statistical Analyses section (page 18, lines 10 – 12) and transferred this to the Discussion section, page 56, line 13.

17.8 Delete (V3…)

RESPONSE: Deleted

17.9 Before this last paragraph, methodology to assess acceptability is missing.

RESPONSE: We have moved the acceptability assessments to just prior to the sample size and statistical analyses section (page 17, lines 15 – 21) to keep the same order of Safety, PK, PD and Acceptability.

17.19 “No participant…. Is a Safety result, no allocation of population.

RESPONSE: We clarified that safety results are reported in the evaluable population. Page 18, lines 16 – 19. 

31.1 No previous mention to “expulsions” in objectives, nor in methods…. Actually, no mention of expulsion in this paragraph beyond the title….

RESPONSE: The expulsions and adherence results section describes the study population’s compliance with the study product. We have therefore moved these results to the study population section, page 24. We also added the data that no participant reported a spontaneous IVR expulsion in the daily study diary.

31.1 If you consider Adherence as part of Acceptability assessment, explain and merge. If you consider as part of follow-up, then must be described before main results: Safety > Pk>Pd>Accept.

RESPONSE: We agree that adherence is part of the follow up of the study population. We have therefore moved these results to the study population section page 24.

45.5 A first paragraph of wrap-up is usually recommended at the beginning of the Discussion.

RESPONSE: We have added a wrap up sentence which summarizes PK and PD data to the beginning of the Discussion, page 50. 

46.9 Cannot considered out of caution. Taking in consideration the occurrence of genital ulcers in this and in Keller’s studies, this AE must be considered as a serious concern to be carefully evaluated within the present and future clinical development. It would probably be considered as a risk by Health Authorities and probably, additional measures of safety surveillance will be required.

RESPONSE: We have now removed this verbage.

46.12-16 The two potential causes of ulcers are here confounded: 1-drug effect over the epithelium; 2-ring’s mechanical effect. COMMENT: both must be investigated in future trials.

RESPONSE: We agree and have cited drug effect (reference 34) and mechanical effect, as we clarified that 90% of patients enrolled in this trial were sexually active.

47.8-9 SERIOUS INCONSISTENCY: TEAE means Treatment – Emergent……therefore, by definition, it is considered related with the treatment !!!!!!!!!!!!!!!!!!!!!!!!!!!!!!

RESPONSE: To add clarity, we have changed the term treatment emergent adverse events (TEAEs) to adverse events (AEs). There are treatment-emergent and treatment-related AEs. We considered TEAEs those AEs occurring during treatment, whether they were related to treatment or not. As noted in the Methods section, adverse events were collected at every visit after enrollment. Each AE was graded as to severity and relatedness to study product or study procedures. Not all AEs that occur during a trial are related to the study product or procedure. All AEs were monitored by an independent data monitor and reported to the FDA as part of the Clinical Study Report. 

We hope that these third round of changes makes our manuscript acceptable for publication in PLoS One.

Sincerely

Andrea R. Thurman MD

1. Thurman AR, Ravel J, Gajer P, Marzinke MA, Ouattara LA, Jacot T, et al. Vaginal Microbiota and Mucosal Pharmacokinetics of Tenofovir in Healthy Women Using a 90-Day Tenofovir/Levonorgestrel Vaginal Ring. Front Cell Infect Microbiol. 2022;12:799501. Epub 20220308. doi: 10.3389/fcimb.2022.799501. PubMed PMID: 35350436; PubMed Central PMCID: PMCPMC8957918.

---

## [Decision Letter · Decision Letter 3]

12 Aug 2022

PONE-D-21-23955R3

Randomized, Placebo Controlled Phase I Trial of the Safety, Pharmacokinetics, Pharmacodynamics and Acceptability of a 90 day Tenofovir Plus Levonorgestrel Vaginal Ring used Continuously or Cyclically in Women

PLOS ONE

Dear Dr. Thurman,

Thank you for submitting your manuscript to PLOS ONE. After careful consideration, we feel that it has merit but does not fully meet PLOS ONE’s publication criteria as it currently stands. Therefore, we invite you to submit a revised version of the manuscript that addresses the points raised during the review process.

Please address the final minor considerations raised by the reviewer.

We look forward to receiving your revised manuscript.

Kind regards,

Vanessa Carels

Staff Editor

PLOS ONE

Journal Requirements:

Reviewers' comments:

Reviewer's Responses to Questions

**Comments to the Author**

1. If the authors have adequately addressed your comments raised in a previous round of review and you feel that this manuscript is now acceptable for publication, you may indicate that here to bypass the “Comments to the Author” section, enter your conflict of interest statement in the “Confidential to Editor” section, and submit your "Accept" recommendation.

Reviewer #2: All comments have been addressed

2. Is the manuscript technically sound, and do the data support the conclusions?

Reviewer #2: Yes

3. Has the statistical analysis been performed appropriately and rigorously? 

Reviewer #2: Yes

4. Have the authors made all data underlying the findings in their manuscript fully available?

Reviewer #2: Yes

5. Is the manuscript presented in an intelligible fashion and written in standard English?

Reviewer #2: Yes

6. Review Comments to the Author

Reviewer #2: Thanks to the authors for this new effort and congratulations for the improvement. The questions were appropriately addressed and the present version, with only minor suggestions would be acceptable.

TITLE: Apologize for adding this suggestion now, but previous topics were of higher priority. Please, consider to add the name of the study at the end of the title: “Randomized …. used Continuously or Cyclically in Women: The CONRAD 138 Study”.

Pg.5, Ln.19: I assume that a contraceptive method was recommended, since sexual activity was allowed and there is a placebo arm. A sentence must be included in this section.

Pg.6, Ln.13: “(BV)” is not necessary, since it is not used later.

10, 15: Please, include the method, algorithm or scale used to determine relationship to AE to study product (as well done in the previous line specifying the DAIDS for severity).

46, 9-10: Sentence unclear, please review.

Pages 50-52: Considering this discussion and having in mind HIV protection objective, would you recommend the continuous or the intermittent use? Or this question is still unclear and must be elucidated in future studies? A sentence in the Conclusions section would be appropriate.

7. PLOS authors have the option to publish the peer review history of their article (what does this mean?). If published, this will include your full peer review and any attached files.

Reviewer #2: **Yes: **J. Algorta, MD, PhD

---

## [Author Response · Author response to Decision Letter 3]

22 Aug 2022

Andrea Thurman, MD

Professor of Obstetrics and Gynecology

Eastern Virginia Medical School/CONRAD

601 Colley Ave, Norfolk, VA 23507

Email: thurmaar@evms.edu, Phone: 210-380-5241

22 AUG 2022

Dear Vanessa Carels, Editorial Team Members and Reviewers,

Thank you for your thoughtful fourth review of our manuscript, PONE-D-21-23955R3, “Randomized, Placebo Controlled Phase I Trial of Safety, Pharmacokinetics, Pharmacodynamics and Acceptability of a 90 day Tenofovir Plus Levonorgestrel Vaginal Ring in Women”. We have addressed all of the additional reviewer’s comments and recommended edits below. We point to the page and line number(s) of the changes in a revised, tracked version of the manuscript. We hope that these changes finally make our manuscript acceptable for publication.

REVIEWER COMMENTS:

Reviewer #2: Thanks to the authors for this new effort and congratulations for the improvement. The questions were appropriately addressed and the present version, with only minor suggestions would be acceptable.

TITLE: Apologize for adding this suggestion now, but previous topics were of higher priority. Please, consider to add the name of the study at the end of the title: “Randomized …. used Continuously or Cyclically in Women: The CONRAD 138 Study”.

RESPONSE: We have added “The CONRAD 138 Study” to the title.

Pg.5, Ln.19: I assume that a contraceptive method was recommended, since sexual activity was allowed and there is a placebo arm. A sentence must be included in this section.

RESPONSE: We included that participants could not be at risk of pregnancy due to the consistent use of condoms, sterilization or heterosexual abstinence on page 6, lines 10 – 12.

Pg.6, Ln.13: “(BV)” is not necessary, since it is not used later.

RESPONSE: This abbreviation has been deleted on page 6, line 15.

10, 15: Please, include the method, algorithm or scale used to determine relationship to AE to study product (as well done in the previous line specifying the DAIDS for severity).

RESPONSE: We clarified on page 10, line 15 that the relationship of the treatment emergent adverse event to study product or study procedures was graded as related or not related. This process included assignment of relatedness by the clinical site PI and confirmation of classification by the Sponsor’s Medical Director. We did not include other sub-divisions of this grading scale such as possibly related, probably related, etc.

46, 9-10: Sentence unclear, please review.

RESPONSE: We have re-written this sentence, now on page 46, lines 9 – 12 to state, “Twelve of 18 TFV/LNG IVR continuous dosing users reported mild or moderate AEs versus 15 of 18 TFV/LNG IVR cyclic dosing users. While this difference was statistically significant, we do not believe that it is clinically relevant, as the difference is small and AEs were mostly mild and unrelated to product use.”

Pages 50-52: Considering this discussion and having in mind HIV protection objective, would you recommend the continuous or the intermittent use? Or this question is still unclear and must be elucidated in future studies? A sentence in the Conclusions section would be appropriate.

RESPONSE: Our data appear to support a preferred recommendation for continuous IVR use, as this regimen shows a more consistent PK/PD profile. We have added clarification to the discussion section of LNG PK (page 49, lines 15 - 20) that although we do not know if the decreases in serum LNG concentration with IVR removal would ultimately impact contraceptive efficacy, the fact that interrupted use does not offer an advantage over continuous use in menstrual bleeding patterns, supports continuous use. We have also added a statement to the conclusion section, page 54, lines 5 - 10 stating that “Because cyclic dosing did not offer an advantage in the menstrual bleeding pattern over continuous use, our data currently would support a preferred recommendation of continuous IVR use, as this regimen shows a more consistent PK/PD pattern. Long-lasting protective levels of TFV-DP in tissue, however, would also support ring removal for short periods.” 

Again, we thank the team for their thoughtful review and hope that this 4th revision makes the manuscript acceptable for publication in PLoS One.

Sincerely,

Andrea Thurman, MD

---

## [Editor Report · Decision Letter 4]

26 Sep 2022

Randomized, Placebo Controlled Phase I Trial of the Safety, Pharmacokinetics, Pharmacodynamics and Acceptability of a 90 day Tenofovir Plus Levonorgestrel Vaginal Ring used Continuously or Cyclically in Women : The CONRAD 138 Study

PONE-D-21-23955R4

Dear Dr. Thurman,

We’re pleased to inform you that your manuscript has been judged scientifically suitable for publication and will be formally accepted for publication once it meets all outstanding technical requirements.

Kind regards,

Vanessa Carels

Staff Editor

PLOS ONE
---

## [Editor Report · Acceptance letter]

28 Sep 2022

PONE-D-21-23955R4 

Randomized, Placebo Controlled Phase I Trial of the Safety, Pharmacokinetics, Pharmacodynamics and Acceptability of a 90 day Tenofovir Plus Levonorgestrel Vaginal Ring used Continuously or Cyclically in Women:  The CONRAD 138 Study 

Dear Dr. Thurman:

I'm pleased to inform you that your manuscript has been deemed suitable for publication in PLOS ONE. Congratulations! Your manuscript is now with our production department. 

Kind regards, 

on behalf of

Dr. Vanessa Carels 

Staff Editor

PLOS ONE